# Seasonal changes in sea ice kinematics and deformation in the Pacific Sector of the Arctic Ocean in 2018/19

Ruibo Lei[1], Mario Hoppmann[2], Bin Cheng[3], Guangyu Zuo[1,4], Dawei Gui[1,5], Qiongqiong Cai[6], H. Jakob Belter[2], Wangxiao Yang[4]

[1] Key Laboratory for Polar Science of the MNR, Polar Research Institute of China, Shanghai, China.

[2] Alfred-Wegener-Institut Helmholtz-Zentrum für Polar- und Meeresforschung, Bremerhaven, Germany.

[3] Finnish Meteorological Institute, Helsinki, Finland.

[4] College of Electrical and Power Engineering, Taiyuan University of Technology, Taiyuan, China.

[5] Chinese Antarctic Center of Surveying and Mapping, Wuhan University, Wuhan, China.

[6] National Marine Environmental Forecasting Center of the MNR, Beijing, China.

*Correspondence to*: Ruibo Lei (leiruibo@pric.org.cn)

**Abstract.** Arctic sea ice kinematics and deformation play significant roles in heat and momentum exchange between the atmosphere and ocean, and at the same time have profound impacts on biological processes and biogeochemical cycles. However, the mechanisms regulating their changes on seasonal scales and their spatial variability remain poorly understood. Using position data recorded by 32 buoys in the Pacific sector of the Arctic Ocean (PAO), we characterized the spatiotemporal variations in ice kinematics and deformation for autumn–winter 2018/19, during the transition from a melting sea ice regime to a near consolidated ice pack. In autumn, the response of the sea ice drift to wind and inertial forcing were stronger in the southern and western PAO compared to the northern and eastern PAO. These spatial heterogeneities gradually weakened from autumn to winter, in line with the increases in ice concentration and thickness. Correspondingly, ice deformation became much more localized as the sea ice mechanical strength increased, with the area proportion occupied by the strongest (15%) ice deformation decreasing by about 50 % from autumn to winter. During the freezing season, ice deformation rate in the northern PAO was about 2.5 times higher than in the western PAO and probably related to the higher spatial heterogeneity of oceanic and atmospheric forcing in the north. North–south and east–west gradients in sea ice kinematics and deformation within the PAO, as observed in this study,

are likely to become more pronounced in the future as a result of a longer melt season, especially in the
western and southern parts.
**1 Introduction**
The Pacific sector of Arctic Ocean (PAO) includes the Beaufort, Chukchi, and East Siberian Seas, as
well as the Canadian and Makarov Basins. Among all the different sectors of the Arctic Ocean, the PAO
exhibited the largest decrease in both seasonal sea ice (Comiso et al., 2017) and multi-year sea ice (MYI)
(Serreze and Meier, 2018) in recent decades. These changes are most likely attributed to an enhanced
ice–albedo feedback (Steele and Dickinson, 2016), increased Pacific water inflow (Woodgate et al.,
2012), and a more pronounced Arctic Dipole (Lei et al., 2016). In the PAO, MYI is mainly distributed
north of the Canadian Arctic Archipelago (Lindell and Long, 2016), suggesting a strong east–west
gradient in sea ice thickness and strength. In summer, the marginal ice zone (MIZ), defined as the area in
which the sea ice concentration is less than 80 %, can reach as far north as 80° N (Strong and Rigor,
2013), thus the south–north gradient in sea ice properties in the PAO is expected to be larger compared to
other sectors of the Arctic Ocean.
Sea ice deformation typically results from the divergence/convergence of ice floes and the presence of
shear stresses, which can enhance redistribution of ice thickness and/or sea ice production by creating
leads and ridges (Hutchings and Hibler, 2008; Itkin et al., 2018). Loss of MYI and a decreased ice
thickness weaken the Arctic sea ice cover, increase floe mobility (Spreen et al., 2011), and promote ice
deformation (Kwok, 2006). Leads forming between ice floes increase heat transfer from the ocean to the
atmosphere, a process that is particularly important in winter because of the large temperature gradient
(Alam and Curry, 1998). In summer, cracks, leads or polynyas within the pack ice represent windows
that expose the ocean to more sunlight. They may significantly alter many biological processes and
biogeochemical cycles, for example supporting under-ice haptophyte algae blooms (Assmy et al., 2017).
Under converging conditions, ice blocks are packed randomly during the formation of pressure ridges,
creating water-filled voids that act as thermal buffers for subsequent ice growth (Salganik et al., 2020).
The high porosity of pressure ridges provides an abundance of nutrients for ice algae communities. As a
result, pressure ridges can become biological hotspots (Fernández-Méndez et al., 2018). Thus, accurate

characterizations of sea ice deformation are not only relevant to a better understanding of ice dynamics and its role in Arctic climate system, but especially also of the evolution of ice-associated ecosystems.

In the PAO, the generally anticyclonic Beaufort Gyre (BG) governs a sea ice motion that is clockwise on average. The boundary and strength of the BG are mainly regulated by the Beaufort High (BH) (Proshutinsky et al., 2009; Lei et al., 2019). An anomalously low BH can result in a reversal of wind and ice motion in the PAO that is normally anticyclonic (Moore et al., 2018). Under a positive Arctic Dipole Anomaly (DA), more sea ice from the PAO is transported to the Atlantic sector of the Arctic Ocean (AAO), i.e. promoting ice advection from the BG system to the Transpolar Drift Stream (TDS) (Wang et al., 2009). In summer, such a regime would stimulate the ice–albedo feedback and accelerate sea ice retreat in the PAO (Lei et al., 2016). The loss of PAO summer sea ice observed during the last four decades can be explained by an increase of ice advection from the PAO to the AAO by 9.6% (Bi et al., 2019). In the zonal direction, the enhanced anticyclonic circulation in the PAO, which is majorly related to a positive BH anomaly (Lei et al., 2019), can result in a larger ice advection from the Beaufort and Chukchi Seas to the East Siberian Sea (Ding et al., 2017). The response of sea ice advection in this region to interannual variations of atmospheric circulation patterns has been studied extensively (e.g., Vihma et al., 2012), but investigations of ice deformation on a seasonal scale are relatively scarce.

From a dynamical perspective, sea ice consolidation has been related to the strength of the inertial signal of sea ice motion (Gimbert et al., 2012), ice–wind speed ratio (IWSR) (Haller et al., 2014), localization, intermittence and space–time coupling of sea ice deformation (Marsan et al., 2004), as well as the response of ice deformation to wind forcing (Haller et al., 2014). The inertial oscillation is caused by the earth's rotation and is stimulated by sudden changes in external forces, mainly due to enhanced wind stress on the ice-ocean interface and surface mixed layer during storms/cyclones or moving fronts of extreme weather events (e.g., Lammert et al., 2009; Gimbert et al., 2012). It usually is weakened by the friction at the ice-ocean interface and internal ice stresses. The localization and intermittence of sea ice deformation indicate the degree of constraint for its spatial range and temporal duration (Rampal et al., 2008). Space-time coupling demonstrates the temporal or spatial dependence of the spatial or temporal scaling laws of ice deformation, which can indicate the brittle behaviour of sea ice deformation (Rampal et al., 2008; Marsan and Weiss, 2010). The inertial oscillations of sea ice motion (Gimbert et

al., 2012) and the IWSR (Spreen et al., 2011) in the Arctic Ocean have been increasingly associated with
reduced sea ice thickness and concentration.
The application of drifting ice buoys to determine the properties and seasonal cycle of the atmosphere,
ocean, and sea ice on a basin scale and year-round has been an emerging technique in polar research in
recent years. For example, drifting buoys are a suitable tool to track relative ice motion. However, the
limited presence of such buoys in a given region and season due to financial and logistical constraints has
made it difficult so far to accurately distinguish spatial variability and temporal changes in sea ice
kinematics and deformation in the PAO. During spring 2003, the deformation of a single lead in the
Beaufort Sea was investigated using Global Positioning System (GPS) receivers (Hutchings and Hibler,
2008). Sea ice deformation and its length scaling law in the southern PAO during March–May have
been estimated before by Hutchings et al. (2011 and 2018) and Itkin et al. (2017). Based on the
dispersion characteristics of ice motion estimated from buoy data recorded in the southern Beaufort Sea,
Lukovich et al. (2011) found that the scaling law of absolute zonal dispersion is about twice that in the
meridional direction. Lei et al. (2020a and 2020b) used data recorded by two buoy arrays deployed in
the northern PAO to describe the influence of cyclonic activities and the summer sea ice regime on the
seasonal evolution of sea ice deformation. In addition to in-situ buoy data, high resolution satellite
images (e.g., Kwok, 2006) and sea ice numerical models (e.g., Hutter et al., 2018) have been used to
identify spatial and temporal variations of ice deformation on a basin scale. RADARSAT data for
example revealed that the length scaling law of ice deformation in the western Arctic Ocean increased
in summer as the ice pack weakens and internal stresses cannot be transmitted over long distances
compared to winter (Stern and Lindsay, 2009). However, an assessment of the ability of satellite
techniques to accurately characterize ice deformation, which often occurs on much smaller scales than
the image resolution and over much shorter periods than their retrieval interval (Hutchings and Hibler,
2008), still requires more ground-truthing data as provided by drifting buoys. So far, a comprehensive
picture of spatial and seasonal variations of sea ice kinematics and deformation for the PAO region has
not yet been obtained, and our understanding is particularly limited with respect to the transition from
the melting season to a near rigid-lid consolidated ice pack in winter.
In order to address the knowledge gaps outlined above, 27 drifting buoys were deployed on sea ice in the
PAO during August and September 2018 by the Chinese National Arctic Research Expedition
(CHINARE) and the TICE expedition led by the Alfred-Wegener-Institute. In this study, we combined

the data measured by these buoys with other available buoy data from the International Arctic Buoy

Programme (IABP) to identify the spatial variability of sea ice kinematics and deformation in the PAO

from melting to freezing season, and linked these results to the atmospheric forcing responsible for the

observed changes in ice dynamics.

**2 Data and Methods**

**2.1 Deployment of drifting buoys**

Four types of buoys were used in this study (Fig. 1): the Snow and Ice Mass Balance Array (SIMBA)

buoy manufactured by the Scottish Association for Marine Science Research Services Ltd, Oban,

Scotland; the Snow Buoy (SB) designed by the Alfred-Wegener-Institute and manufactured by

MetOcean Telematics, Halifax, Canada; the ice Surface Velocity Program drifting buoy (iSVP) also

manufactured by MetOcean Telematics; and the ice drifter manufactured by the Taiyuan University of

Technology (TUT), China. All buoys were equipped with GPS receivers providing a positioning

accuracy of better than 5 m and regularly reporting to a land-based receiving system using the Iridium

satellite network.

During the CHINARE, 9 SIMBA buoys and 11 TUT buoys were deployed in a narrow zonal section of

156° –171° W and a wide meridional range of 79.2° – 84.9° N in August 2018 (Figs. 1 and 2). This

deployment scheme was designed to facilitate the analysis of changes in ice kinematics from the loose

MIZ to the consolidated Pack Ice Zone (PIZ). Of these 20 buoys, 15 were deployed in the northern part

of the PAO as a cluster within close distance of each other (black circles in Fig. 2) to allow an estimation

of ice deformation rates. In addition, data from five SIMBAs and two SBs deployed by the TICE

expedition in the Makarov Basin during September 2018 (Figs. 1 and 2) were also used to estimate ice

deformation rates. Because the ice thickness at the deployment sites was comparably large (1.22 to 2.49

m), the buoys were able to survive into winter and beyond. Position data from one iSVP deployed during

the previous CHINARE in 2016 (Lei et al., 2020a) and four other IABP buoys were also included in this

study. The IABP buoys were deployed by the British Antarctic Survey and Environment Canada in the

east of the PAO during August –September 2018. Here we use the position data from these 32 buoys to

describe spatial variations in ice kinematics (Fig. 2) between August 2018 and February 2019. We chose

this study period because it represents a transition period during which the mechanical properties of sea

ice are expected to change considerably (e.g., Herman and Glowacki, 2012; Hutter et al., 2018).
Two-thirds of them (22) continued to send data until or beyond the end of the study period. During this
study period, the buoy trajectories during the study period roughly covered the region of 76° – 87° N
and 155° E – 110°W, which we define here as our study region.
**2.2 Analysis of sea ice kinematic characteristics**
All buoys were configured to a sampling interval of either 0.5 or 1 h. Prior to the calculation of ice
drift velocity, position data measured by the buoys were interpolated to a regular interval ($\tau$) of 1 h. To
quantify meridional (zonal) variabilities of ice kinematic properties, we used data from buoys that
were within one standard deviation of the average longitude (latitude). This constraint helped to
minimize the influence of the zonal (meridional) difference on the meridional (zonal) variabilities.
The resulting meridional extent for the assessment of the zonal variabilities of ice kinematics ranged
from 350 to 402 km, while the zonal extent for the assessment of the meridional variabilities ranged
from 195 to 285 km. Their seasonal changes can be considered as moderate (<40%) although a
divergence of the floes occurred at all times. Using half a standard deviation to constrain the
calculation range, there is no essential change in the identified meridional/zonal dependencies of ice
kinematics from those obtained using one standard deviation. Thus, we consider our evaluation
method as robust. Meridional variabilities are related to the transition from the MIZ to the PIZ, while
zonal variabilities indicate the change between the region north of the Canadian Arctic Archipelago,
where MYI coverage is usually large (Lindell and Long, 2016) and the Makarov Basin, which is
mainly covered by seasonal ice (Serreze and Meier, 2018).
Two parameters were used to characterize sea ice kinematics. First, the IWSR was used to investigate
the response of the sea ice motion to wind forcing. Impacts of data resampling intervals (1–48 h),
meridional and zonal spatial variabilities, intensity of wind forcing, near-surface air temperature, and
ice concentration on the IWSR were assessed. These parameters are either related to spatiotemporal
changes in atmospheric and sea ice conditions, or to the frequency characteristics of ice and wind
speeds. The data used to characterize the atmospheric forcing, including sea level air pressure (SLP),
near-surface air temperature at 2 m ($T_{2m}$) and wind velocity at 10 m ($W_{10m}$), were obtained from the
ECMWF ERA-Interim reanalysis dataset (Dee et al., 2011). Sea ice concentration was obtained from
the Advanced Microwave Scanning Radiometer 2 (AMSR2) (Spreen et al., 2008). To identify the state
of the atmospheric forcing and the sea ice conditions relative to the climatology, we also calculated
anomalies of SLP, $T_{2m}$, $W_{10m}$, ice concentration, and ice drift speed relative to the 1979–2018
averages. To estimate ice concentration anomalies, we used ice concentration data from the Nimbus-7
Scanning Multichannel Microwave Radiometer (SMMR) and its successors (SSM/I and SSMIS)
(Fetterer et al., 2017) because they cover a longer period compared to the AMSR2 data. We used the
daily product of sea ice motion (Tschudi et al., 2019 and 2020) provided by the National Snow and Ice
Data Center (NSIDC) to estimate anomalies of ice speed. However, this could be only estimated for
August–December 2018 because of the delayed release of NSIDC data.
Second, the inertial motion index (IMI) was used to quantify the inertial component of the ice motion.
To obtain the IMI, we applied a Fast Fourier Transformation to normalized hourly ice velocities.
Normalized ice velocities were calculated by scaling the velocity values to monthly averages,
allowing seasonal change to be assessed independently of the magnitudes of ice velocities. The
frequency of the inertial oscillation varies with latitude according to
$$f_0 = 2\Omega \sin\theta , \tag{1}$$
where $f_0$ is the inertial frequency, $\Omega$ is the Earth rotation rate, and $\theta$ is the latitude. $f_0$ ranges from 2.01
to 1.94 cycles day$^{-1}$ between 90° N and 75° N. Rotary spectra calculated from sea ice velocity using
complex Fourier analysis were used to identify signals of inertial or tidal origin, both of which have a
frequency of ~ 2 cycles day$^{-1}$ in the Arctic Ocean (Gimbert et al., 2012). According to Gimbert et al.
(2012), the complex Fourier transformation $\widehat{U}(\omega)$ is defined as:
$$\widehat{U}(\omega) = \frac{1}{N} \sum_{t=t_0}^{t_{end}-\Delta t} e^{-i\omega t} \left( u_x + i u_y \right), \tag{2}$$
where $N$ and $\Delta t$ are the number and temporal interval of velocity samples, $t_0$ and $t_{end}$ are the start and end
times of the temporal window, $u_x$ and $u_y$ are the zonal and meridional ice speeds at $t+0.5\Delta t$ on an
orthogonal geographical grid, and $\omega$ is the angular frequency. The IMI is defined as the amplitude at the
negative-phase inertial frequency, i.e., $-f_0$, after the complex Fourier transformation. The energies that
contributed to the amplitude at $-f_0$ comprise the potential contributions from quasi-semidiurnal inertial
and tidal oscillations, as well as the high-frequency components of wind and oceanic forcing; while
those in the positive phase, $f_0$, excludes contributions from the inertial oscillation and only comprises
other components compared to that at –$f_0$. This is because the spectral peaks associated with the tidal
oscillation are roughly symmetric at positive and negative phases as a first order approximation
(Gimbert et al., 2012). On the contrary, the spectral peak associated with the inertial oscillation is
asymmetric and only occurs in the negative phase in the Arctic Ocean. Thus, we will identify the
seasonal changes in the contributions of the inertial oscillation by comparing the amplitude at the
negative-phase quasi-semidiurnal frequency, i.e., IMI, to the positive-phase amplitude (PHA). Such
method to separate the inertial oscillation from the tidal oscillation has been used by Lammert et al.
(2009), who attempted to identify cyclone-induced inertial ice oscillation in Fram Strait. The
background noise originating from high-frequency components of wind and oceanic forcing can
slightly shift the local maxima from the targeted frequencies of the IMI and PHA (Geiger and Perovich,
2008). Thus, we identify the local maximum amplitude in the range of –$f_0\pm0.03$ for the IMI and in the
range of 2±0.03 for the PHA. If no local maximum can be identified within the predefined ranges, we
use the amplitudes at –$f_0$ and 2 as the IMI and PHA, respectively. Such a situation is encountered in 15%
of the IMI cases, and in 95% of the PHA cases. This implies that the inertial oscillation is much more
prevalent, while the tidal oscillation can be ignored regardless of seasons and buoys under
consideration. This result might be related to the fact that, throughout the study period, all the buoys
drifted over the deep basins far beyond the continental shelf.
**2.3 Analysis of sea ice deformation characteristics**
Buoy position data were also used to estimate differential kinematic properties (DKPs) of the sea ice
deformation field. The DKPs include divergence (*div*), shear (*shr*), and total deformation (*D*) rates of
sea ice estimated within the area enclosed by any three buoys, as shown by Itkin et al. (2017).
Following Hutchings and Hibler (2008), DKPs were calculated as follows:
$$div = \frac{\partial u}{\partial x} + \frac{\partial v}{\partial y} \ , \tag{3}$$
$$shr = \sqrt{\left(\frac{\partial u}{\partial x} - \frac{\partial v}{\partial y}\right)^2 + \left(\frac{\partial u}{\partial y} + \frac{\partial v}{\partial x}\right)^2} \ , \tag{4}$$
and $D= \sqrt{div^2 + shr^2}$, $\tag{5}$
where $\frac{\partial u}{\partial x}, \frac{\partial u}{\partial y}, \ \frac{\partial v}{\partial x}$, and $\frac{\partial v}{\partial y}$ are the strain components on an orthogonal geographical grid. Sea ice strain
rate was only estimated for those buoy triangles with internal angles in excess of 15° and for ice speeds
larger than 0.02 m s$^{-1}$ to ensure a high accuracy (Hutchings et al., 2012). Total deformation *D* was used
to characterize the spatial and temporal scaling laws as follows:
$D \propto L^{-\beta}$, (6)
and $D \propto \tau^{-\alpha}$, (7)
where $L$ is the length scale, $\tau$ is the sampling interval, and $\beta$ and $\alpha$ are spatial and temporal scaling
exponents which indicate the decay rates of ice deformation in the spatial or temporal domains. These
scaling laws can only indicate the fractal properties of the first moment of ice deformation because of
the multi-fractal properties of ice deformation (e.g., Marsan et al., 2004; Hutchings et al., 2011 and
2018). To estimate the spatial exponent $\beta$ for the CHINARE buoy cluster, the length scale was divided
into three bins of 5–10, 10–20, and 20–40 km because only few samples were outside these bins. To
estimate the temporal exponent $\alpha$, the position data were resampled to intervals of 1, 2, 4, 8, 12, 24,
and 48 h. Because the TICE buoy cluster was mostly (> 70 %) assigned to the 40–80 km bin, data
from this cluster were not suitable for the estimation of the scale effect. A space–time coupling index,
$c$, denoting temporal (spatial) dependence of the spatial (temporal) scaling exponent, can be expressed
as:
$\beta(\tau) = \beta_0 - c\ln(\tau)$, (8)
where $\beta_0$ is a constant. The areal localization index, $\delta_{15\%}$, was used to quantify the localization of the
strongest sea ice deformation, defined as the fractional area accommodating the largest 15 % of the ice
deformation in the research domain (Stern and Lindsay, 2009). The $\delta_{15\%}$ was calculated for the 10–20
km length bin for the CHINARE buoy cluster, since this bin contained more samples to ensure a
statistical rationality. To identify the influence of the temporal scale on the localization of ice
deformation, all data were resampled to intervals of 1, 2, 4, 8, 12, 24, and 48 h.
**2.4 Atmospheric circulation pattern**
We calculated the seasonal Central Arctic Index (CAI) and DA index to relate these large-scale
atmospheric circulation patterns to the potential of sea ice advection from the study region to the AAO
(Vihma et al., 2012; Bi et al., 2019). Further, we calculated the seasonal AO and BH indices to relate
them to the strength of the BG (Lei et al., 2019). Monthly SLP data north of 70° N obtained from the
NCEP/NCAR reanalysis I dataset were used to calculate the empirical orthogonal functions (EOF), with
the AO and DA as the first and second modes of the EOF (Wang et al., 2009). The CAI was defined as
the difference in SLP between 90° W and 90° E at 84° N (Vihma et al., 2012). The BH index was
calculated as the SLP anomaly over the domain of 75°–85° N, 170° E–150° W (Moore et al., 2018)
relative to 1979–2018 climatology.
**3 Results and discussions**
**3.1 Spatial and seasonal changes in atmospheric and sea ice conditions**
The BH index for autumn (September, October, and November) 2018 was moderate, ranking the tenth
highest in 1979–2018 (Fig. 3a). However, the BH index for the following winter (December, January,
and February) was much lower at −5.6 hPa, ranking the fourth lowest in 1979–2018 (Fig. 3b). Both,
CAI and DA, were positive in autumn 2018, but still within one standard deviation of the 1979–2018
climatology (Fig. 3c and 3e). However, both indices were strongly positive in winter 2018/19, ranking
the third and second highest in 1979–2018, respectively (Fig. 3d and 3f). The sea ice in the PAO is
expected to be considerably impacted by these seasonal changes in atmospheric circulation patterns as
a result of the enhanced northward advection of sea ice to the AAO (e.g., Bi et al., 2019). As an
example, a pronounced sea ice reduction has been observed in the Bering Sea in March 2019, where
sea ice extent was 70 %–80 % lower than normal (Perovich et al., 2019).
Associated with the seasonal change in the BH index, there was a distinct contrast in the pattern of the
BG from anticyclonic in autumn to cyclonic in winter. Wind vectors and ice drift trajectories during
autumn 2018 were generally clockwise, while those during the following winter were counterclockwise.
The latter resulted in all buoys drifting northeastward and integrating into the TDS from December
2018 onward (Fig. 4). In autumn 2018, strong northerly winds only appeared in the northwestern part
of study region (Fig. 4a), and were associated with a moderately positive CAI and DA. However, in
winter 2018/2019, enhanced northerly winds prevailed almost across the entire study region (Fig. 4b),
and were associated with an extremely positive CAI and DA. The $T_{2m}$ anomalies averaged over the
study region were 3.9 °C in autumn and 0.7 °C in winter (Fig. 4c and 4d), ranking the second and
eleventh highest in 1979–2018, respectively.
The CHINARE buoys were deployed within a narrow meridional section at about 170° W. On 20
August 2018, sea ice concentration in this section, and especially in the southern part, was considerably
lower than that in the eastern part of the study region at about 120° W, where other buoys had been
deployed (Fig. 5a). Subsequently, ice concentration increased considerably, with almost all buoys being
located in the PIZ by 20 September 2018 (Fig. 5b). However, the CHINARE buoys in the south and all
TICE buoys remained within 70 km from the ice edge, which retreated further during August–
September 2018. By 20 October 2018, ice concentration surrounding all buoys had increased to over
95 % (Fig. 5c).
In September and early October 2018, ice concentrations were considerably lower than the 1979–2018
average. Ice concentrations increased after early October and became comparable with climatological
values (Figs. 6b and 7b). In October 2018, ice concentration was much lower in the southern and
western parts of the study region compared to the north and east. Subsequently, the spatial gradient of
sea ice concentration gradually decreased. Compared to the 1979–2018 climatology, wind speed was
lower throughout most of the study period except for episodic increases as a result of intrusions of
low-pressure systems (Figs. 6c and 7c). In September 2018, ice speed in the south was higher
compared to the north (Fig. 6d), suggesting that the sea ice response to wind forcing was stronger in the
south because of the lower ice concentration. From October 2018 onwards, this north–south difference
gradually disappeared. The study region was dominated by a low SLP during December 2018 and
February 2019, which was related to an anomalously low BH index and subsequent increases in both
wind and ice drift speeds (Figs. 6c, 6d, 7c, and 7d).

**3.2 Spatial and seasonal changes in sea ice kinematic characteristics**

Temporal resampling has little effect on wind speed. However, applying longer resampling intervals to
buoy position data may filter out ice motions that occur at higher frequencies (Haller et al., 2014),
resulting in reduced ice speed and IWSR (Fig. 8). For example, ice drift speed and IWSR in September
2018 were 0.13 m s$^{-1}$ and 0.027 at a resampling interval of 1 h, and decreased to 0.01 m s$^{-1}$ and 0.021
at a resampling interval of 48 h. Both ice speed and IWSR decreased considerably from September to
November 2018; afterwards, both variables remained low until the end of the study period. At a
resampling interval of 6 h, the IWSR was 0.026 in September 2018 (Fig. 8), which is much lower than
that (0.013) obtained in the region close to North Pole in the same month of 2007 (Haller et al., 2014)
because most parts of our study region included the MIZ at that time. This value decreased to 0.008–
0.015 during November to February (Fig. 8), which is comparable to those obtained from the regions
north of Siberia or Greenland and the region close to North Pole during the freezing season, but much
smaller than that obtained in Fram Strait (Haller et al., 2014). This implies that, during the freezing

season, the response of the sea ice to wind forcing is relatively uniform for the entire Arctic Ocean

except for the regions close to Fram Strait where ice speeds markedly increases. In January 2019, a

more consolidated ice pack and a relatively weak wind forcing led to both ice drift speed and IWSR

reaching minima for the entire study period (Figs. 6c and 7c). The influence of resampling on the

IWSR was reduced considerably during the freezing season, implying significant reductions of

meandering and sub-daily oscillations in ice motion compared to the melt season. The ratio between

IWSRs at 1-h and 48-h intervals in October was 70 % of that in September and remained almost

unchanged between November and February.

Factors regulating the IWSR are summarized in Table 1. The impact of the geographical location was

significant in autumn, with relatively high IWSRs in the southern or western parts of the study region.

However, meridional changes in the IWSR became very small in January–February because the north–

south gradient in ice conditions was negligible by that time. The west–east gradient was more

pronounced, with a significant relationship between longitude and IWSR throughout the study period.

This is consistent with the results given by Lukovich et al. (2011), who identified that the west–east

gradient of sea ice motion is larger than that in the north–south direction for the southern PAO during

the freezing season. In summer and early autumn, the consolidation of the ice field is low, and

interactions between individual ice floes approximate rigid particle collisions (Lewis and

Richter-Menge, 1998). Thus, in August–October 2018, a lower IWSR is related to stronger wind

forcing that enhanced the interactions between floes, which leads to a significant negative statistical

correlation between the IWSR and wind speed. Similarly, based on the data obtained from the buoys

deployed in the TDS region, Haller et al. (2014) also identified that some spikes of the IWSR tend to be

associated with a low wind speed. Consolidation of the ice field between November and February 2018

led to reduced ice motion and weaker sea ice response to wind forcing. Thereby, impact of wind

forcing on IWSR was insignificant from November onwards. Variations of $T_{2m}$ across the study region

between 20 August and 30 September 2018 were relatively small (−1.7 to −3.5 °C) because of the

thermodynamic balance between the sea ice and the atmosphere during the melt season (e.g., Screen

and Simmonds, 2010). The statistical relationship between $T_{2m}$ and the IWSR was insignificant during

this period. However, the relationship became significant during October–December 2018, with a

higher $T_{2m}$ being associated with a larger IWSR because warmer conditions may have weakened ice

pack (e.g., Oikkonen et al., 2017). As the continuing thickening of the ice cover further reduced the

influence of air temperature on ice kinematics, the statistical relationship between $T_{2m}$ and the IWSR was insignificant in January and February 2019.

The initial strength of the inertial oscillation mainly depends on the wind stress. However, the sustainability of the inertial oscillation is restricted by the internal friction within the Ekman layer in regions with low ice concentration and much open water, or by the ice internal stress in the PIZ (Gimbert et al., 2012). Thus, the inertial component of ice motion is closely associated with the seasonal and spatial changes in ice conditions. Figure 9 shows monthly IMI and PHA obtained from each buoy displayed at the midpoint of the buoy's trajectory for different months. The combined average IMI of all buoys was 0.099 ± 0.088 for the entire study period, with the average for September 2018 (0.227) being considerably higher. Combined monthly average IMIs from all buoys decreased from 0.136 in October 2018 to 0.037 in February 2019. Spatial variability of the IMI had almost disappeared by February 2019; the IMI standard deviation in February 2019 was 13 %–22 % of that in September–October 2018. Both the magnitude and the spatiotemporal variations of the PHA were much smaller than those of the IMI. The combined average PHA of all available buoys during the entire study period was only 18% of the IMI. The monthly ratio between the PHA and IMI ranged from 0.06 in September 2018 to 0.46 in February 2019. The seasonal damping of this ratio is mainly due to the decrease in the IMI because no statistically significant trend can be identified for the PHA. The standard deviation of the IMI revealed a significant decreasing trend ($P<0.01$) from 0.069–0.117 in September–October 2018 to 0.015 in February 2019, which suggests that the spatial variation of the IMI gradually decreased as the winter approached. Similar to the ratio between the absolute magnitudes, the ratio between the standard deviations of the PHA and IMI increased from 0.08 in September to 0.50–0.70 in January–February. The seasonal increase in this ratio also was mainly due to the decrease in the standard deviation of the IMI. From comparisons between the seasonalities of the IMI and PHA, we infer that the seasonal changes and spatial variations in the IMI could be mainly related to the changes in the inertial oscillation, and the contributions of the tidal oscillation can be ignored throughout the study period.

To eliminate the influence of large-scale spatial variability, we inspected subsets of data obtained from the buoys that were deployed in clusters. The IMI obtained from the CHINARE buoy cluster (black circles in Fig. 2) decreased markedly from 0.223 to 0.081 during September–October 2018. However, a similar change was observed one month later in October–November 2018 for the TICE buoy cluster.

During the freezing season from November to February, the IMI gradually decreased to 0.038 for the
CHINARE cluster and to 0.035 for the TICE cluster. Sea ice growth rates of the thin ice in the MIZ in
the western and southern PAO are expected to be higher than that in the PIZ in the northern and the
eastern PAO (e.g., Kwok and Cunningham, 2008). Accordingly, the spatial variability of the ice inertial
oscillation observed in early autumn gradually disappeared.
To study the temporal changes in the IMI and PHA in more detail, we used a complex Fourier
transformation to obtain time series of the IMI and PHA based on a 5-day temporal window. Here, we
only show selected results from three representative buoys for comparison (Fig. 10). Those buoys were
initially located in the southernmost and northernmost domain of the CHINARE cluster, and in the
southernmost domain of the TICE cluster (Fig. 2). The timing of the distinct seasonal attenuation of the
IMI was different between the buoys, occurring in mid-October, late September, and late October 2018
for the CHINARE southernmost and northernmost buoys, and the TICE southernmost buoy,
respectively (Fig. 10). During the freezing season, the IMI remained at a low level, but was still always
larger than the PHA. The magnitude of the IMI was mainly regulated by wind forcing during the
freezing season. The wind speed can significantly explain the magnitude of the IMI in November–
February by 22% ($P<0.05$), 45% ($P<0.001$), and 21% ($P<0.05$) for the CHINARE southernmost and
northernmost buoys, and the TICE southernmost buoy, respectively. The relatively large wind speed is
related to a relatively low IMI because the enhanced wind forcing might increase the ice internal stress
and reduce the response of ice motion to inertia forcing. This mechanism is most obvious in the
northern PIZ because of the relatively large ice internal stress.
**3.3 Spatial and seasonal changes in sea ice deformation**
For all buoy triangles that were used to estimate ice deformation, the ice concentration within the
CHINARE buoy cluster increased most rapidly during late August and early September 2018, and it
remained close to 100 % from then onwards (Fig. 11a). A comparable seasonal increase in ice
concentration was observed within the TICE buoy cluster one month later. To facilitate a direct
comparison of the data obtained by the two different buoy clusters, we estimated the ice deformation
rate of the TICE buoy cluster at the 10–20 km scale using the value at the 40–80 km scale and a
constant spatial scaling exponent of 0.55. The scaling exponent of 0.55 is a seasonal average obtained
from the CHINARE buoy cluster. A change of the scaling exponent by 10 % would lead to an
uncertainty of about 0.03 for the ice deformation rate. Thus, a constant scaling exponent can be
considered acceptable for a study of seasonal variation. In early and mid-September 2018, the ice
deformation rate was low for the CHINARE cluster (Fig. 11b) because of low and relatively stable
wind forcing (Fig. 2). For the TICE cluster, both ice deformation rate and ratio between ice
deformation rate and wind speed decreased rapidly between 20 September and 10 November 2018,
associated with a consolidation of the ice field as ice concentration and thickness increased, and
temperature decreased. However, the ice deformation rate obtained by the CHINARE buoy cluster
decreased only slightly over the same period, which is likely linked to its relatively low initial
deformation rate in late September 2018 and to the higher ice concentration (15 %–20 %) compared to
the TICE region.
For the CHINARE buoy cluster, the daily wind speed can explain 35 % ($P<0.001$) of the daily ice
deformation rate estimated from hourly position data throughout the study period. However, for the
TICE cluster, changes in ice deformation were mainly regulated by the seasonal evolution of ice
concentration between September and early November 2018. The relationship between ice deformation
rate and wind speed was insignificant at the statistical confidence level of 0.05 during this period. The
ice field had sufficiently consolidated by mid-November 2018, and the relationship between daily ice
deformation rate and daily wind speed changed to significant ($R^2 = 0.12$, $P<0.01$) from then onwards.
The average ratio of ice deformation rate to wind speed in autumn was $1.15 \times 10^{-6}$ m$^{-1}$ for the
CHINARE cluster and $0.62 \times 10^{-6}$ m$^{-1}$ for the TICE cluster; the ratio in winter decreased to $0.86 \times 10^{-6}$
and $0.17 \times 10^{-6}$ m$^{-1}$, respectively. This seasonal pattern is consistent with results of Spreen et al. (2017),
who used the RGPS data to reveal that the annual maximum ice deformation rate occurred in August,
and decreased gradually to the annual minimum in March. Except for late September 2018, when the
ice concentration in the TICE cluster was less than 85 %, the ice deformation rate from the CHINARE
cluster was generally larger than that of the TICE cluster, with average values of 0.45 and 0.13 d$^{-1}$,
respectively, for October 2018 to February 2019. Sea ice in the region of the TICE cluster was
generally thinner compared to the region of the CHINARE cluster. Thus, the difference in ice
deformation rate cannot be explained by a difference in ice conditions between the two regions, and is
most likely related to the spatial heterogeneity of wind and/or oceanic forcing. Changes in the direction
of wind vectors were more frequent around the CHINARE cluster than around the TICE cluster.
Frequent changes in ice drift direction lead to larger ice deformation events, such as those on 11
October, and 11 and 26 November 2018 for the CHINARE cluster as shown in Fig. 11b. The drifting
trajectories of the TICE cluster were much straighter than those of the CHINARE cluster. Since the
CHINARE cluster was located in the core region of the BG, the vorticity of the surface current must be
greater than that in the TICE cluster, located at the western boundary of the BG (Armitage et al., 2017).
As a result, ice deformation rate and its ratio to wind speed were lower for the TICE cluster.
Ice deformation rates obtained from the CHINARE buoy cluster at three representative lengths of 7.5,
15, and 30 km were estimated using Eq. (6). Figure 12 shows that the monthly average ice deformation
decreased as the length scale and resampling interval increased, implying an ice deformation
localization and intermittency. The ice deformation decreased rapidly at all spatial and temporal scales
during the seasonal transition period of September–October, and remained low from then onwards. Ice
deformation rate obtained using hourly position data from the CHINARE buoy cluster in September
2018 was 0.38 $d^{-1}$ at the length scale of 30 km, which is comparable to that in September 2016 (0.31
$d^{-1}$), and much larger than that in September 2014 (0.18 $d^{-1}$) observed also in northern PAO (Lei et al.,
2020b). These observed differences can be related to the strong storms in late September 2018 (Fig.
11b) and early September 2016 (Lei et al., 2020b), in contrast to the relatively stable synoptic
conditions and relatively compact ice conditions in September 2014 (Lei et al., 2020b).
Accordingly, the spatial scaling exponent $\beta$ estimated from hourly position data was 0.61 in September
2018, which is comparable to $\beta$ from September 2016 (0.60), but slightly larger than in September
2014 (0.46) observed in northern PAO (Lei et al., 2020b). $\beta$ decreased markedly from September to
October 2018, and varied little from then onwards (Fig. 13). With increasing in ice thickness and
concentration as well as a cooling of the ice cover from October onwards, the consolidation of the ice
field is enhanced, and sea ice deformation can spread over longer distances. By February 2019, $\beta$
obtained from hourly position data decreased to 0.48, which is comparable to February 2015 (0.43) in
the northern PAO (Lei et al., 2020a). This suggests that the interannual changes in the spatial scaling of
ice deformation during winter are not as strong as that in early autumn, which is in line with the
evolution of ice thickness (e.g., Kwok and Cunningham, 2008). $\beta$ decreased exponentially with an
increase in resampling frequency for all months, which indicates that the spatial scaling would
generally be underestimated when using data of coarser. Interpolated to 3 h, $\beta$ was 0.42 and 0.44 in
January and February 2019, respectively, which is comparable with the result (0.40) obtained from the
southern PAO during March–May (Itkin et al., 2017). The ice growth season generally lasts until May–
June in the PAO (Perovich et al., 2003), which implies that the sea ice consolidation in March–May is
comparable to, or even stronger than, that in January–February. Thus, our $\beta$ is essentially consistent
with that given by Itkin et al. (2017). Extrapolated to 48 h (120 h), $\beta$ decreased to 0.29 (0.25) in
January and 0.33 (0.28) in February 2019, respectively, which is comparable to that (0.20) obtained
from the estimations using RADARSAT images with temporal resolution of 48–120 h during the
freezing season for the pan-Arctic Ocean (Stern and Linday, 2009). We further use the seasonal bin to
test the sensitivity of the estimation of $\beta$ to the number of samples. Consequentially, the seasonal $\beta$ was
estimated at 0.54 and 0.48 for autumn and winter, respectively, which is close to those (0.53 and 0.49)
averaged directly from the monthly values. Therefore, we believe that the monthly segmentation for
estimations of $\beta$ is statistically appropriate and can better reveal seasonal changes.
The temporal scaling exponent $\alpha$ also exhibited a strong dependence on the spatial scale, which means
a relatively large intermittency of ice deformation can be obtained by fine-scale observations (Fig. 14).
Seasonally, the value of $\alpha$ decreased between September and October 2018 because of enhanced
consolidation of the ice cover. The value of the space–time coupling coefficient $c$ increased
monotonously from 0.034 in autumn to 0.062 in winter, suggesting a gradual enhancement of the brittle
rheology of the ice cover. This is consistent with the results derived from RADARSAT images (Stern
and Moritz, 2002), which revealed that sea ice deformation is more linear in winter, and more clustered
and spatially random in summer. The value of $c$ in September 2018 is comparable to that in September
2016 (0.03). However, it is only about half that in September 2014 (0.06) (Lei et al., 2020b) because of
the different ice conditions. The value of $c$ in January–February 2019 (0.059–0.062) is comparable with
the values obtained in January–February 2015 (0.051–0.077) from the northern PAO (Lei et al.,
2020a), and the value obtained from the region north of Svalbard in winter and spring (Oikkonen et al.,

490    2017).

The areal localization index denotes the area with the highest (15%) deformation. It had a strong
dependence on the temporal scale and increased linearly ($P<0.001$) as the logarithm of the temporal
scale increased (Fig. 15), which implies that the localization of ice deformation would be
underestimated when using coarser temporal resolution. Seasonally, the areal localization index
decreased significantly from September to November 2018, indicating that ice deformation was
increasingly localized during the transition from melting to freezing. In the freezing season, ice
deformation mainly occurs along linear cracks, leads, and/or ridges, which corresponds to a high
localization. During melt season, the ice deforming zones are in clumps rather than along lines. The
spatial distribution of ice deformation rate is more even and amorphous (Stern and Moritz, 2002),
which corresponds to a low localization. From November to February, the degree of deformation
strongly regulated the localization of ice deformation, with the monthly ice deformation rate explaining
96 % of the monthly areal localization index ($P$<0.01). This means that an extremely high ice
deformation can spread over longer distances. The areal localization index for January–February 2019
corresponding to a temporal resolution of 1 h and a length scale of 10–20 km was 1.9 %–2.3 %. This is
close to values estimated using RADARSAT images at a scale of 13–20 km (1.6%) (Marsan et al.,
2004) and at a scale of 10 km (1.5%) (Stern and Lindsay, 2009), as well as that estimated at a scale of
18 km using a high resolution numerical model (2.4 %–2.7 %) (Spreen et al., 2017). We also analyzed
other fractional areas accommodating the largest 10 % or 20 % of the ice deformation. Although the
adjusted indices would have different magnitudes, their overall seasonal patterns and dependence on
the temporal scale are consistent with those using the threshold of 15%. We therefore conclude that the
understanding of the ice deformation localization derived from this study is not very sensitive to the
selected threshold.
**3.4 Spatial differences in the trends of sea ice loss in the PAO and their implications for sea ice**
**kinematics and deformation**
Sea ice conditions in the melt season have profound effects on sea ice dynamic and thermodynamic
processes in the following winters. For example, enhanced divergence of summer sea ice leads to
increased absorption of solar radiation by the upper ocean and delays onset of ice growth (e.g., Lei et
al., 2020b). As shown in Fig. 16, the long-term decrease of sea ice concentration in the first half of
September, when Arctic sea ice extent typically reaches its annual minimum (Comiso et al., 2017), is
stronger in the southern and western PAO than in the northern and eastern PAO. The southern and
western PAO have become ice free in September during recent years. On the contrary, there is no
significant trend in ice concentration in the first half of September along the trajectory of the
easternmost buoy (Fig. 16e). This suggests that the melting period is getting longer in the southern and
western PAO compared to the northern and eastern PAO. Consequently, the spatial gradient of ice
thickness in the PAO, especially during autumn and early winter, will be further enhanced by the delay
in sea ice freezing onset in the south and west. A deformation of the ice field in the seasonal ice zone
creates unfrozen ice ridges (Salganik et al., 2020). These new ridges, together with the newly formed
thin ice in leads, are mechanically vulnerable components of the ice field during the freezing season,
and predispose the ice field to further deformation under external forces. At the end of the freezing
season, the enhanced ice deformation will promote the sea ice breaking up and expand the MIZ
northward, which is conducive to the advance of the melt season. Thus, the north–south and east–west
differences in sea ice kinematics are likely to be more pronounced in the future.
**4 Conclusion and outlook**
High-resolution position data recorded by 32 ice-based drifting buoys in the PAO between August
2018 and February 2019 were analyzed in detail to characterize spatiotemporal variations of sea ice
kinematic and deformation properties. During the transition from autumn to winter, ice deformation
and its response to wind forcing, as well as the inertial signal of ice motion gradually weakened. At the
same time, space–time coupling of ice deformation was enhanced as the mechanical strength of the ice
field increased. The influence of tidal forcing on the quasi-semidiurnal oscillation of ice motion was
negligible regardless of the seasons because the buoys drifted over the deep basins beyond the
continental shelf. During the freezing season between October 2018 and February 2019, the ice
deformation rate in the northern part of the study region was about 2.5 higher compared to in the
western part. This difference is likely related to the higher spatial heterogeneity of the oceanic and
atmospheric forcing in the northern part of the study region, which is situated in the core region of the
BG. Because of the seasonal change in the large-scale atmospheric circulation pattern, indicated by the
enhanced positive phases of the CAI and DA, a significant change in ice drift direction from
anticyclonic to cyclonic patterns was observed in late November 2018, leading to temporal increases in
both ice deformation rate and its ratio to wind forcing.
The pronounced high intermittence of ice deformation suggests that an episodic opening or closing of
the sea ice cover may be undetectable from data with longer sampling intervals, such as remote sensing
data with resolutions of one or two days. Consequently, fluxes of heat (e.g., Heil and Hibler, 2002) or
particles and gases (e.g., Held et al., 2011) released from these openings in the PIZ into the atmosphere
would be underestimated if they are derived from such data. The dependence of the ratio of ice speed to
wind speed on resampling frequency also suggests that the temporal resolution should be considered
carefully when using reanalyzed wind data to parameterize or simulate sea ice drift. From a spatial
perspective, our results reveal that ice deformation intermittence is underestimated at longer scales.
This is consistent with results from numerical models, which indicate that the most extreme
deformation events may be absent in the output of models with lower spatial resolution (Rampal et al.,
2019). This emphasizes the need for high-resolution sea ice dynamic models (e.g., Hutter and Losch,
2020) to reproduce linear kinematic features of ice deformation.
The response of ice kinematics to wind and inertia forcing was stronger in the south and west compared
to the north and east of the study region, which is partly associated with the spatial heterogeneity of ice
conditions inherited from previous seasons. During the transition from autumn to winter, the north–
south and east–west gradients in the IWSR and the inertial component of ice motion gradually
decreased and even disappeared entirely, which is in line with the seasonal evolution of ice
concentration and thickness. The spatial heterogeneity in autumn ice conditions is likely to be
amplified with an increased loss of summer sea ice cover in the southern and western PAO, which is
expected to further enhance the east-west and north-south differences in sea ice kinematics.
We conclude this study by highlighting some of the most important knowledge gaps related to sea ice
kinematics and deformation in the Arctic Ocean, not necessarily limited to the PAO, and how they can
be addressed in the future. First, the spatio-temporal scale effects of ice deformation in this study were
derived based on data recorded by buoys distributed over spatial scales of only 5–40 km. In order to
assess whether the results of the present study are also representative for a much larger domain,
observations by a much wider and denser buoy array, ideally combined with high-resolution ship-based
radar and satellite remote sensing data, as well as the support of numerical models, are needed. Second,
we only examined atmospheric influences on sea ice kinematics and deformation. The ocean also plays
an important role on ice drift and deformation, especially on mesoscales, greatly enhancing ice motion
nonuniformity and ice deformation (e.g., Zhang et al., 1999). In the PAO, mesoscale eddies prevail
over the shelf break and the Northwind and Alpha-Mendeleyev Ridges (e.g., Zhang et al., 1999, Zhao
et al., 2016). To assess the influence of mesoscale oceanic eddies on ice deformation, observations
from ice-drifter arrays are insufficient, highlighting the need for a complementary deployment of
ocean-profiler arrays. Third, deformation of sea ice creates ample opportunity for increased sea ice
biological activities. Irradiance and nutrients, the two major limiting agents for biological growth in the
sea ice realm (Ackley and Sullivan, 1994), are strongly impacted by sea ice deformation. For example,
pressure ridges generally have large semi-enclosed chambers, which can provide more nutrients for
biological activity (Ackley and Sullivan, 1994; Geiger and Perovich, et al., 2008). Sea ice deformation
would also increase ice surface roughness, which in turn increases the potential of melt pond formation
in early summer (e.g., Perovich and Polashenski, 2012). The formation of ponds leads to an increase in
the transmission of irradiance through the ice cover and promote the biological growth (e.g., Nicolaus
et al., 2012). In order to better understand the linkages between sea ice dynamical and biological
processes, more joint observations are urgently needed.
In September 2019, the international Multidisciplinary drifting Observatory for the Study of Arctic
Climate (MOSAiC) drift experiment (2019–2020) was launched in the region north of the Laptev Sea
(Krumpen et al., 2020), which is to the west of the deployment region of the TICE buoy cluster. The
ice thickness around the MOSAiC ice station was much lower than that in the areas of the buoy clusters
included in this study (Krumpen et al., 2020). Frequent sea ice breakup events have been reported
around the MOSAiC ice camp during the drift. An integral part of MOSAiC was the deployment of a
large distributed network of ice-based drifting buoys of various types around the main ice camp.
Supported by a wealth of multi-disciplinary in-situ data, satellite remote sensing data and numerical
model setups, MOSAiC has the potential to properly address all the aspects outlined above. At the
same time, data and results from the present study can be used as a proxy baseline for comparing and
investigating deformation of the MOSAiC ice pack. By comparing our results to the observations from
the MOSAiC buoy array, we may get a broader understanding of the spatial variation of Arctic sea ice
deformation.

**Author contributions**
RL is responsible for project coordination and paper writing. MH, BC, GZ, and GD undertook the
processing and analysis of the buoy data, and interpretation of results. RL, WY, and JB deployed the
buoys. The buoy data were provided by RL, MH, and BC. The atmospheric circulation index was
calculated by QC. All authors commented on the manuscript.

## Data availability

The CHINARE buoy data are archived in the National Arctic and Antarctic Data Centre of China at
https://www.chinare.org.cn/metadata/53de02c5-4524-4be4-b7bb-b56386f1341c (DOI:
10.11856/NNS.D.2020.038.v0). The TICE buoy data are available for download in the online sea-ice
knowledge and data platform www.meereisportal.de and will be archived in PANGAEA. The IABP
buoy data are archived at http://iabp.apl.washington.edu/index.html.

## Competing interests

The authors declare that they have no conflict of interest.

## Acknowledgments

We are most grateful to the Chinese Arctic and Antarctic Administration and the
Alfred-Wegener-Institute for their logistical and financial support of the cruises of CHINARE and
TICE, respectively. We thank the captains, crews and science parties of the R/V Xuelong and the
Akademik Tryoshnikov, especially cruise leaders Dr. Zexun Wei and Dr. Benjamin Rabe, for their
incredible support during the expeditions. The AMSR2 passive microwave ice concentrations were
provided by the University of Bremen. The SMMR-SSMIS ice concentration and ice motion products,
and the monthly Arctic Sea Ice Index were provided by the NSIDC. The ERA-Interim reanalysis was
obtained from the ECMWF. Monthly sea level pressure is obtained from the NCEP/NCAR reanalysis I
dataset. We are very grateful to the two anonymous reviewers and the responsible editor Dr. Ted
Maksym for their comments, which have greatly improved our paper.

## Financial support

This work was supported by grants from the National Key Research and Development Program
(2016YFC1400303, 2018YFA0605903, and 2016YFC1401800) and the National Natural Science
Foundation of China (41722605 and 41976219). B.C. was supported by the European Union's Horizon
2020 research and innovation programme (727890 – INTAROS) and Academy of Finland under contract
317999. The buoys deployed on TICE were funded by the Alfred-Wegener-Institute through the
infrastructure programs FRAM and ACROSS.

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

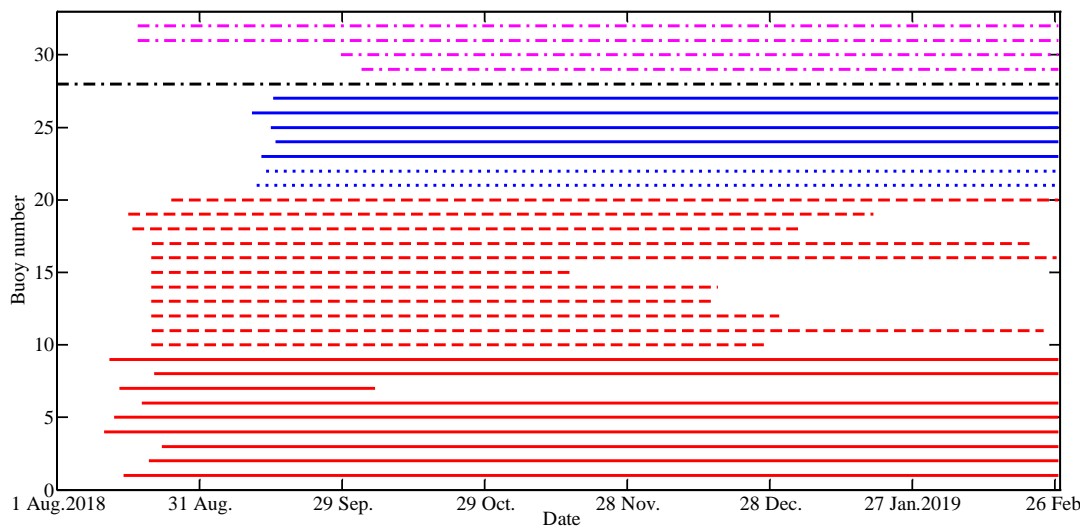


**Figure 1 Operational periods of all buoys included in this study. Red lines denote buoys deployed during CHINARE in August 2018; blue lines denote buoys deployed during TICE; black line indicates the buoy deployed during CHINARE 2016; purple lines represent IABP buoys. Solid, dashed, short-dashed, and dot-dashed lines denote SIMBA, TUT, SB, and iSVP or other buoys, respectively.**

819

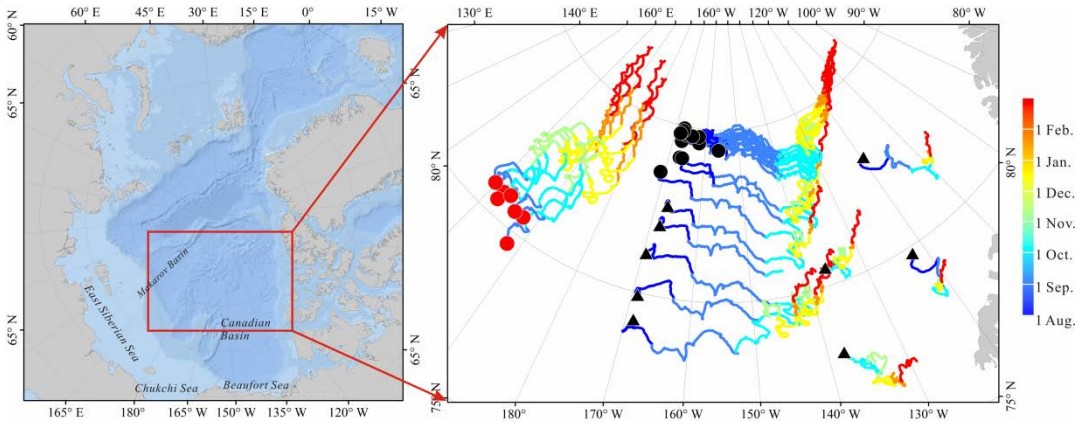

820

**Figure 2 Buoy trajectories between deployment sites (indicated by circles and triangles) and buoy locations on 28 February 2019. Trajectories from 15 buoys deployed during CHINARE at locations indicated by black circles and 7 buoys deployed during TICE at locations indicated by red circles were used to estimate ice deformation rate. For buoys deployed prior to August 2018, the starting point of the trajectory was set to 1 August 2018.**

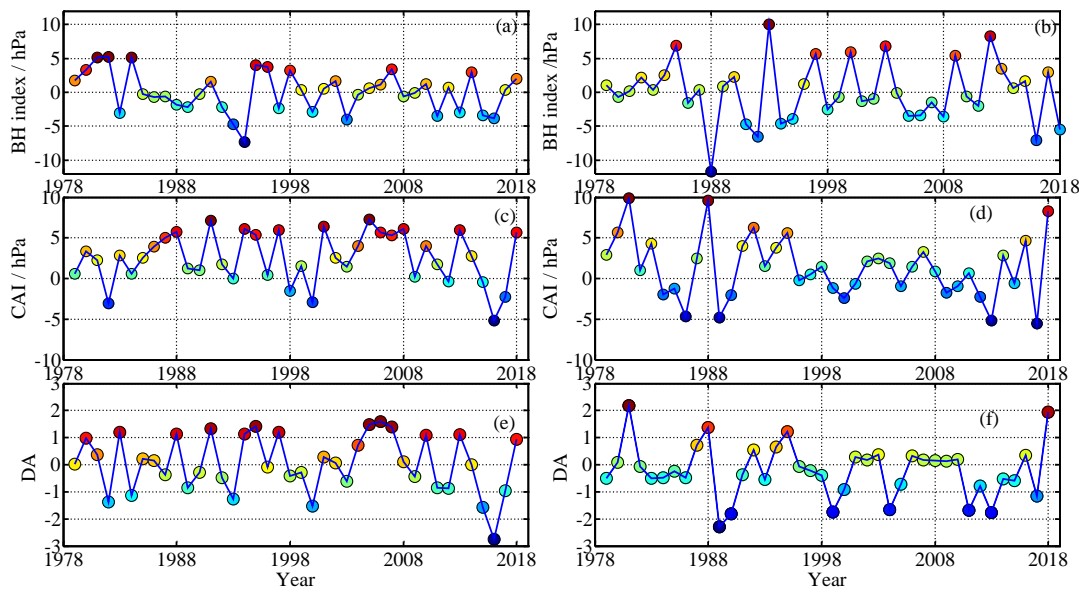


**Figure 3 Changes in (a) autumn (SON) and (b) winter (DJF) BH index, (c) autumn and (d) winter CAI, and (e)**
**autumn and (f) winter DA from 1979 to 2018.**

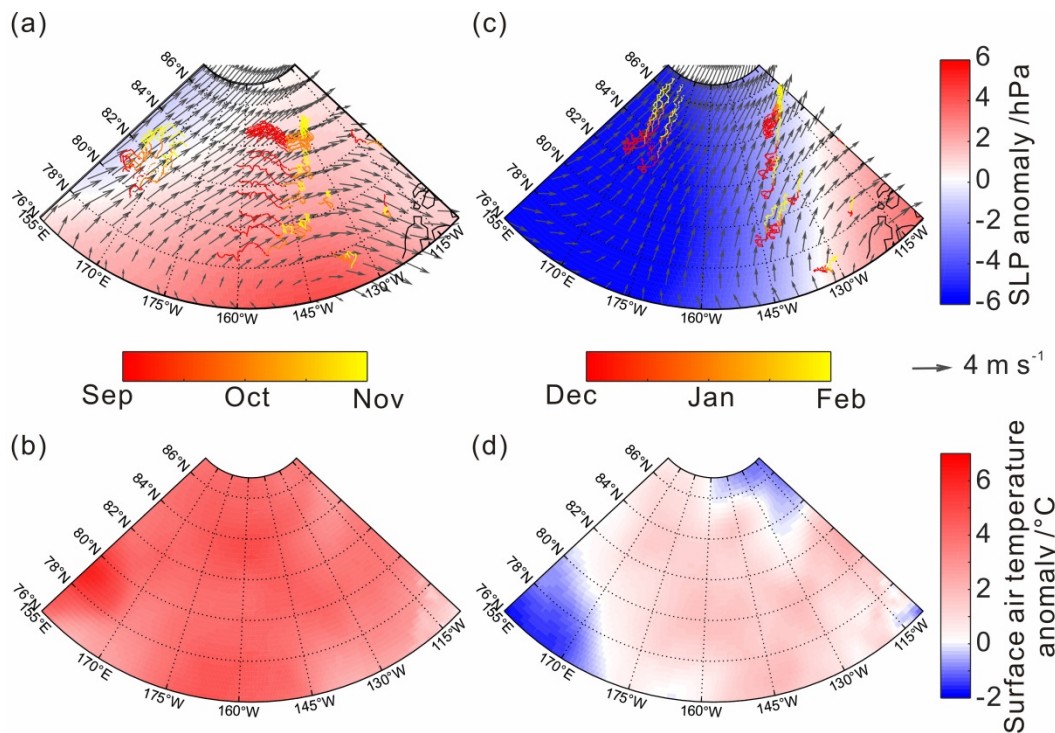


**Figure 4 Anomalies of (a and c) SLP and (b and d) near-surface air temperature (2 m) over the PAO during**
**(a and b) autumn 2018 and (c and d) winter 2018/19 relative to 1979–2018 climatology; (a and c) arrows**
**indicate seasonal average wind vectors and colored lines indicate buoy trajectories through time.**


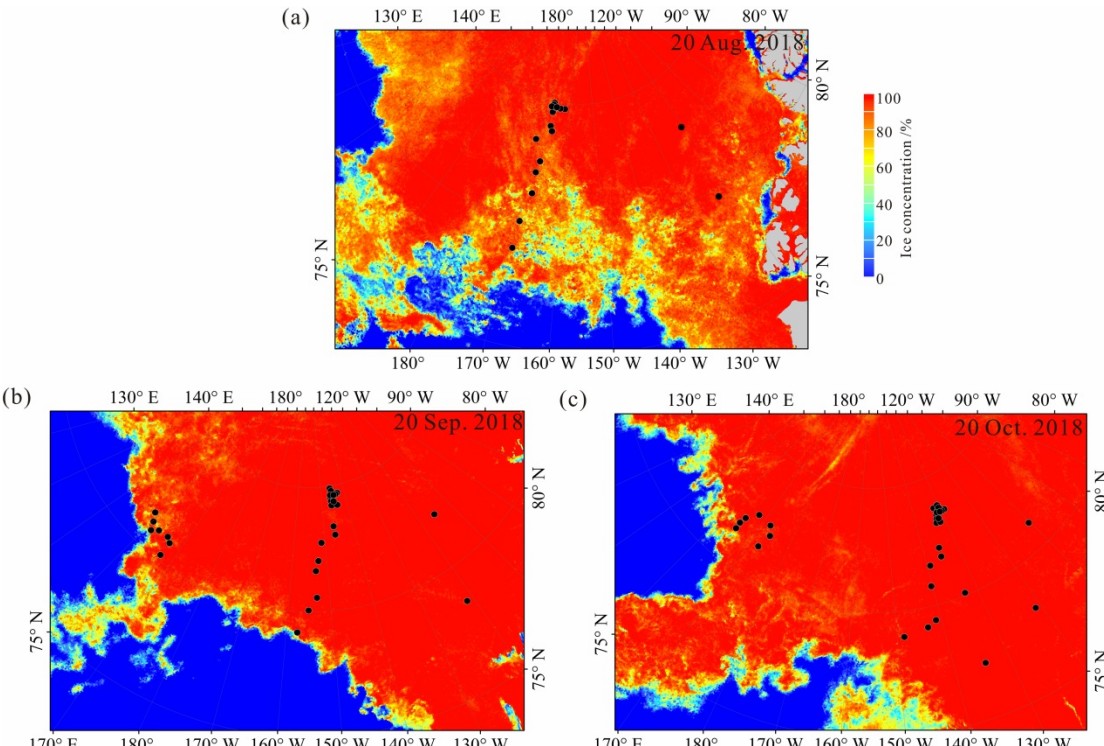

**Figure 5 Sea ice concentration across the PAO on 20 of (a) August, (b) September, and (c) October, 2018, with**
**black dots denoting buoy positions on the given days.**

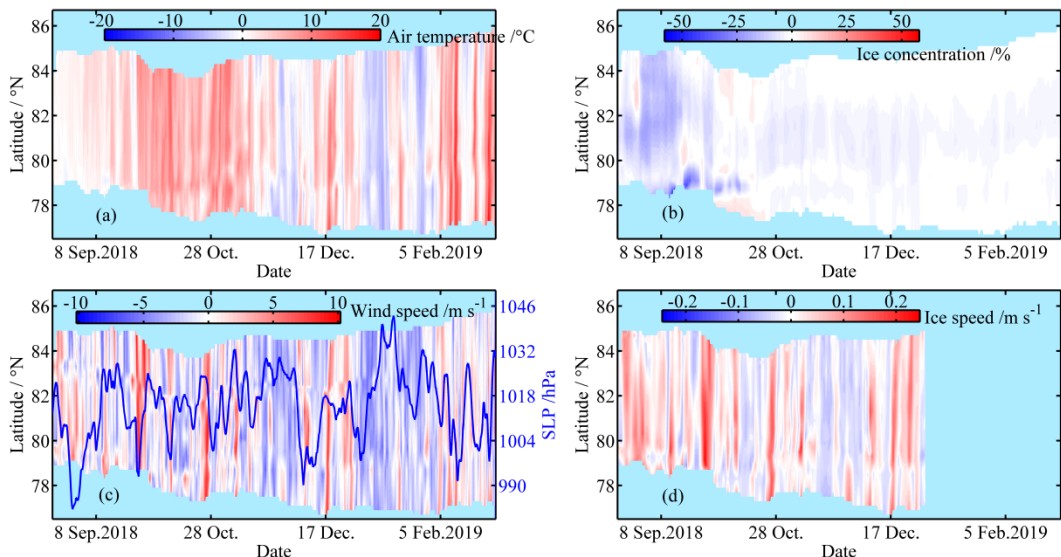

**Figure 6 Meridional and temporal changes in anomalies of (a) $T_{2m}$, (b) ice concentration, (c) wind speed, (d)**
**ice speed in the ice season 2018/19 relative to 1979–2018 climatology; (c) blue line indicates SLP averaged**
**over the study region.**

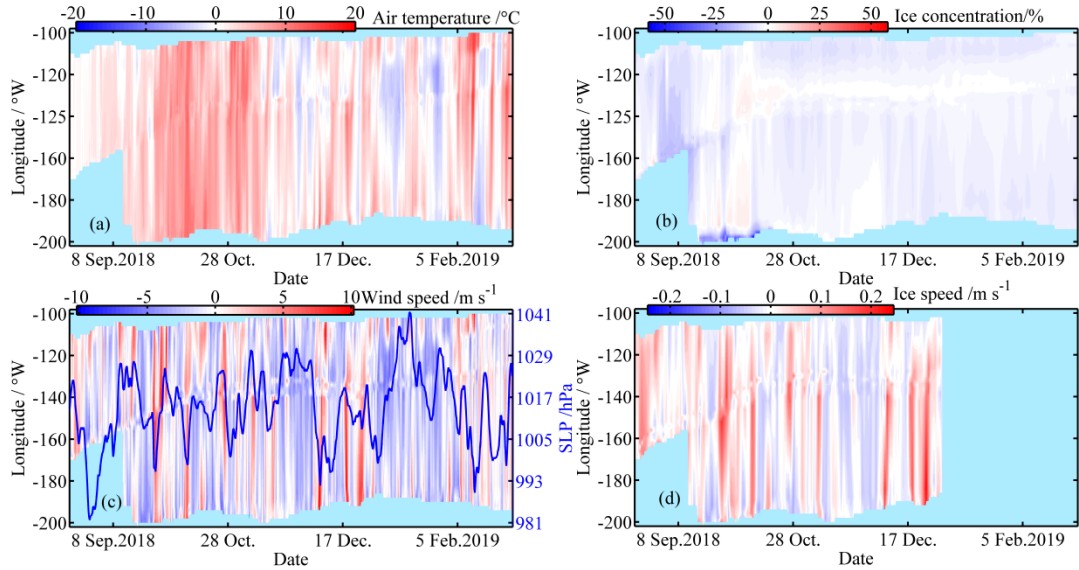


**Figure 7 Same as Fig 2, but for zonal changes. Longitudes with values below −180 denote the eastern Arctic.**

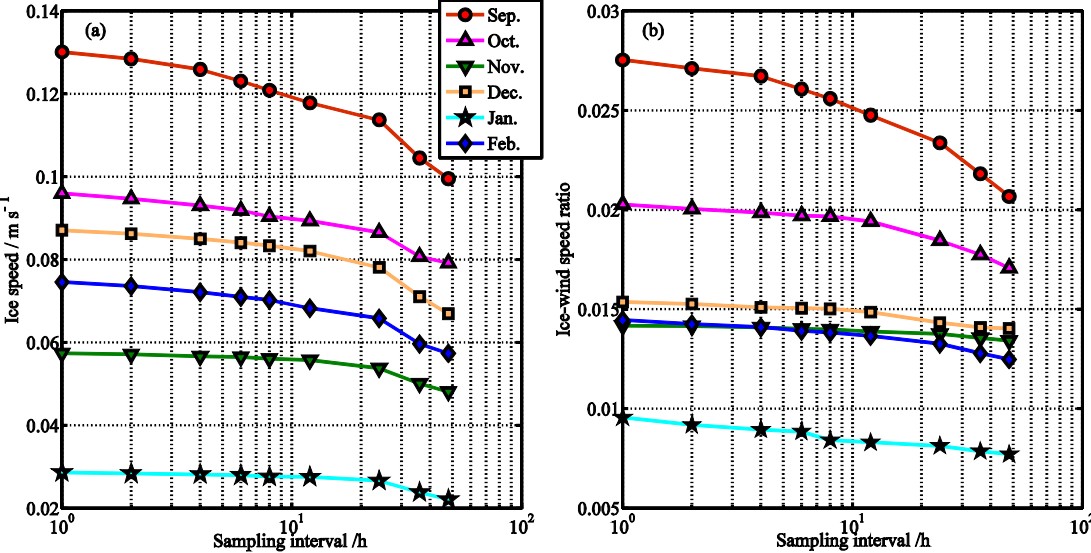


**Figure 8 Changes in (a) ice speed and (b) IWSR as a function of position data resampling interval for**
**various months in 2018/19.**

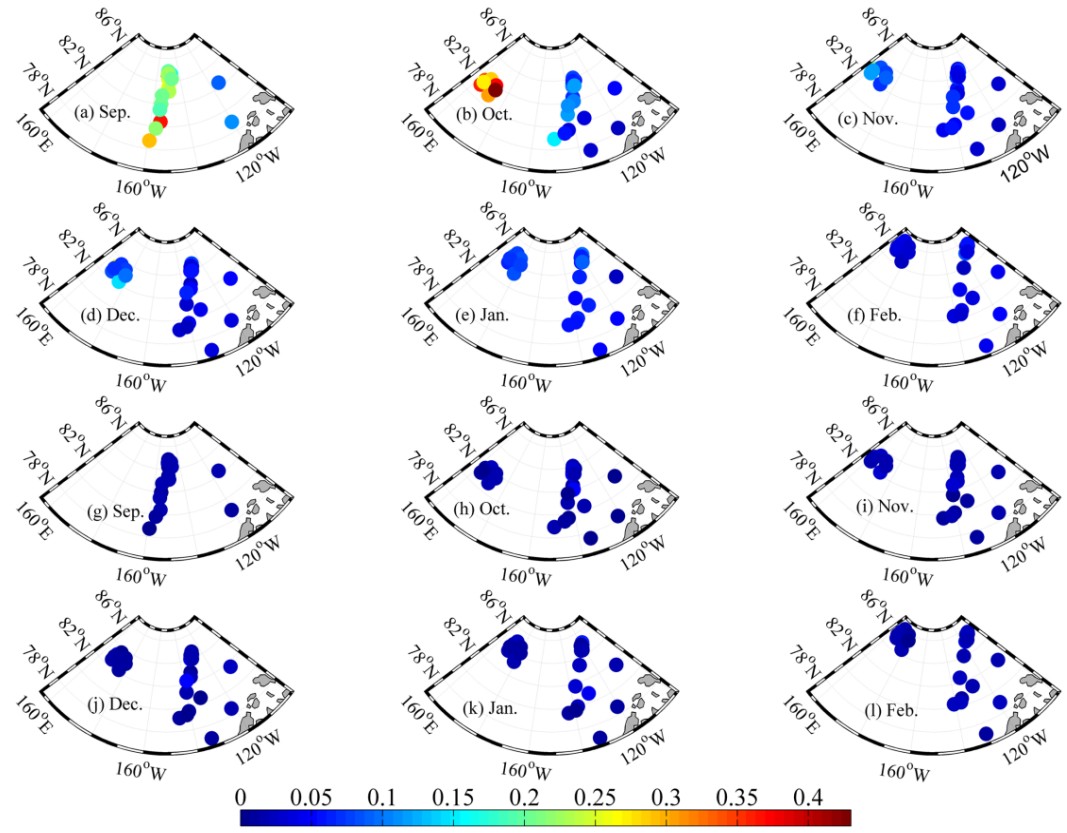


**Figure 9 Amplitudes after Fourier transformation of monthly time series of normalized ice velocity at the negative-phase inertial frequency (a–f) and positive-phase semidiurnal frequency (g–l) from September 2018 to February 2019.**

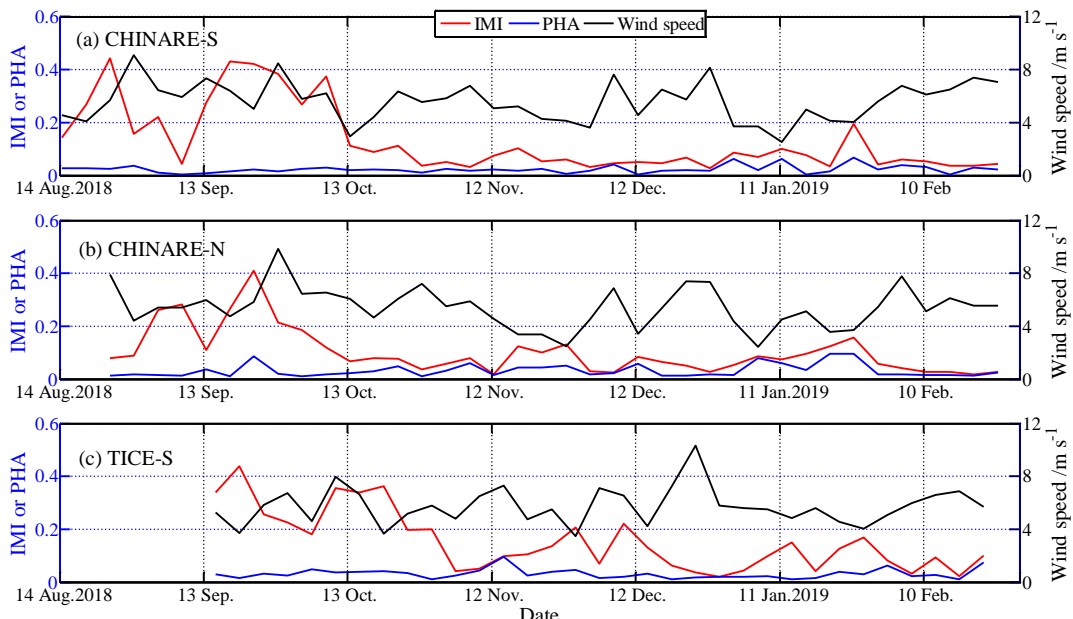


**Figure 10 Amplitudes after Fourier transformation of normalized ice velocity at the negative-phase inertial frequency (IMI) and positive-phase semidiurnal frequency (PHA) obtained from the 5-day temporal window, as well as the corresponding wind speed.**



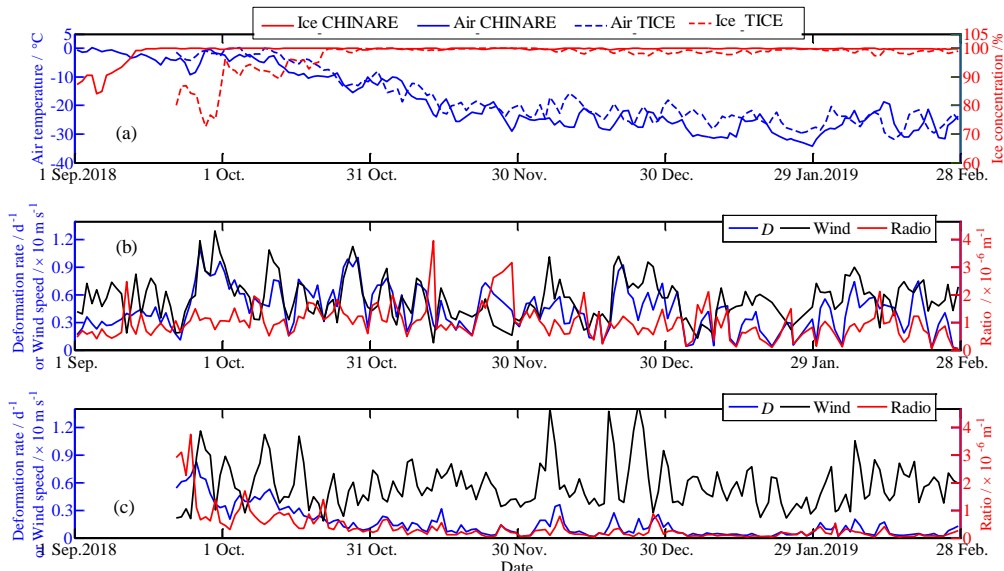


**Figure 11 (a) Time series of daily average near-surface (2 m) air temperature and ice concentration within**
**the CHINARE and TICE buoy clusters. Ice deformation rate (*D*), wind speed and their ratio at the 10–20 km**
**scale for the (b) CHINARE and (c) TICE buoy clusters.**

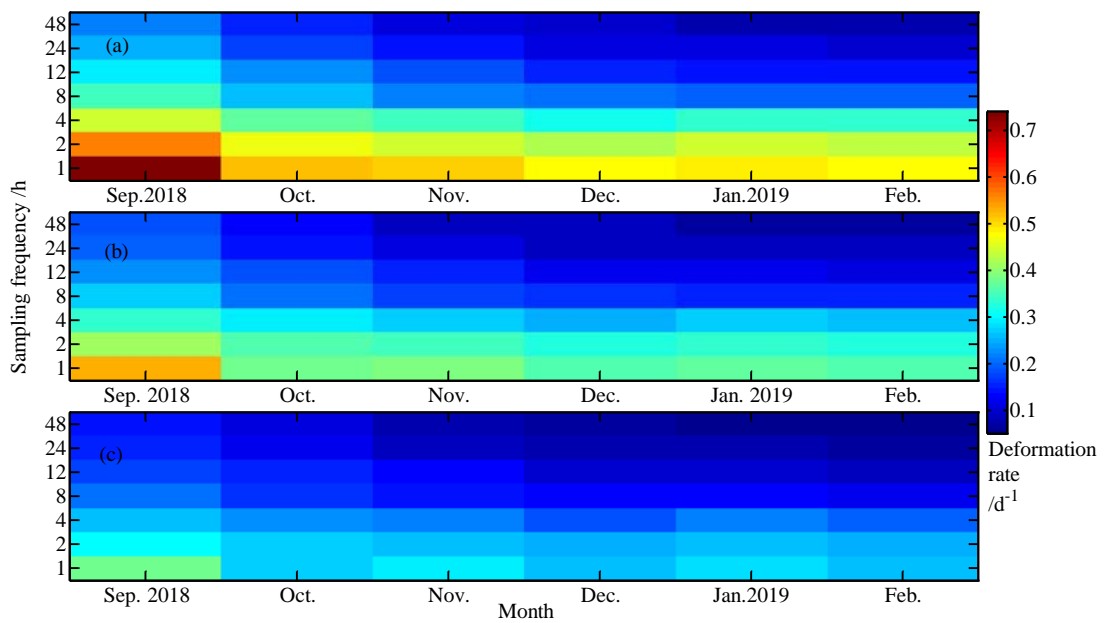


**Figure 12 Monthly average sea ice deformation rate calculated from the CHINARE buoy cluster at length**
**scales of (a) 7.5 km, (b) 15 km, and (c) 30 km using position data resampled at various intervals between 1 and**
**48 h.**

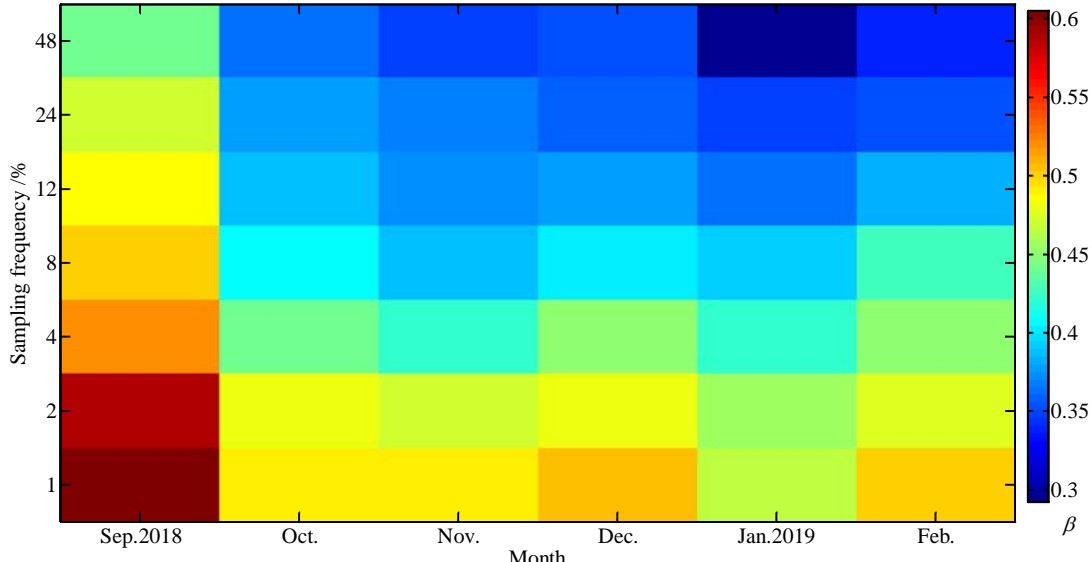


**Figure 13 Changes in monthly spatial scaling exponent as a function of position data resampling frequency**

**obtained from the CHINARE buoy cluster.**


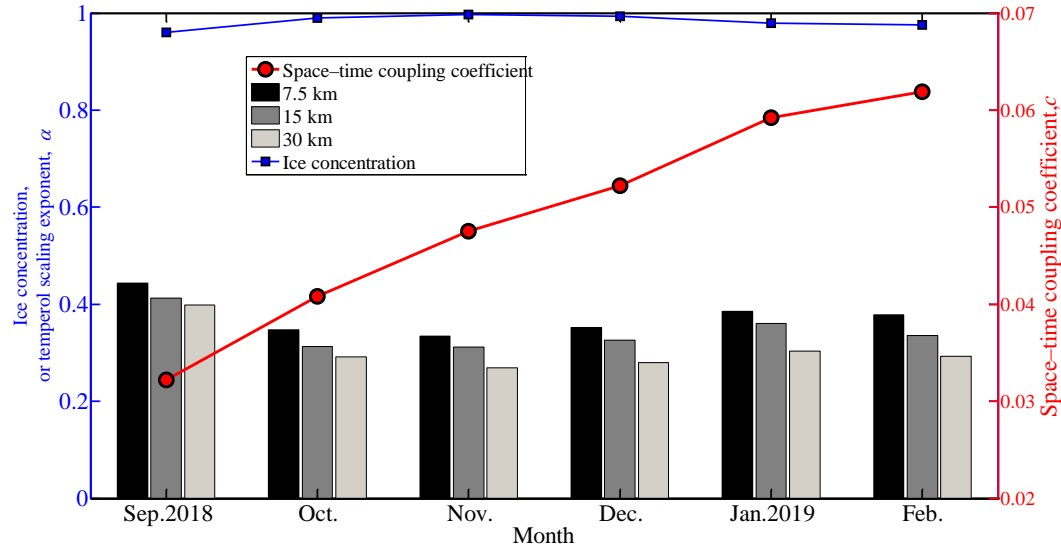


**Figure 14 Changes in monthly temporal scaling exponent at various length scales, space–time coupling**

**coefficient, and average ice concentration within the CHINARE buoy cluster.**

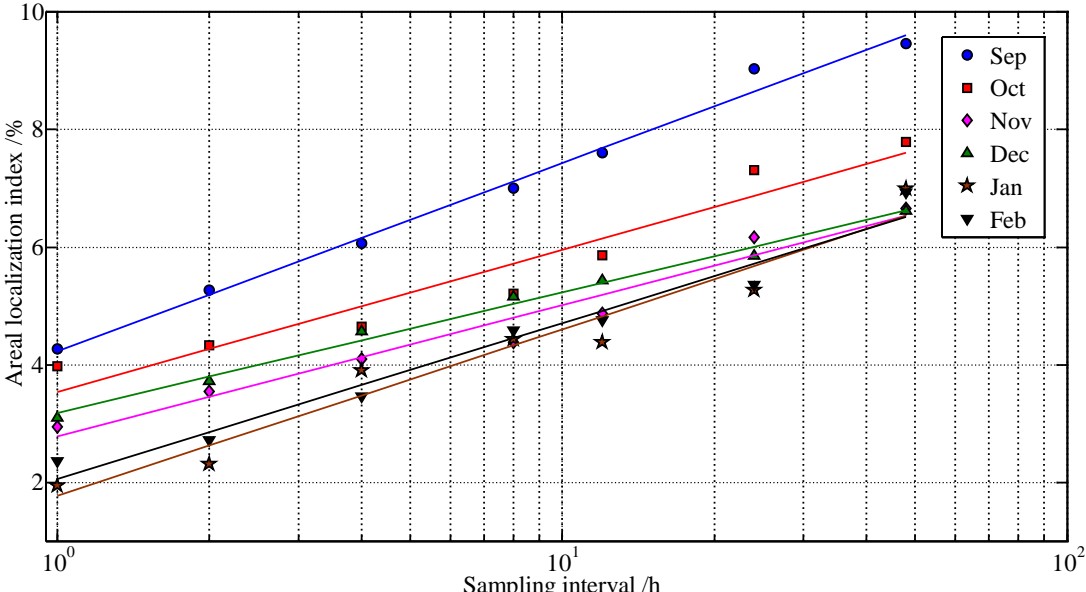


**Figure 15 Changes in monthly (September 2018 to February 2019) areal localization index of ice deformation**

**at a scale of 10–20 km as a function of the position data resampling frequency.**


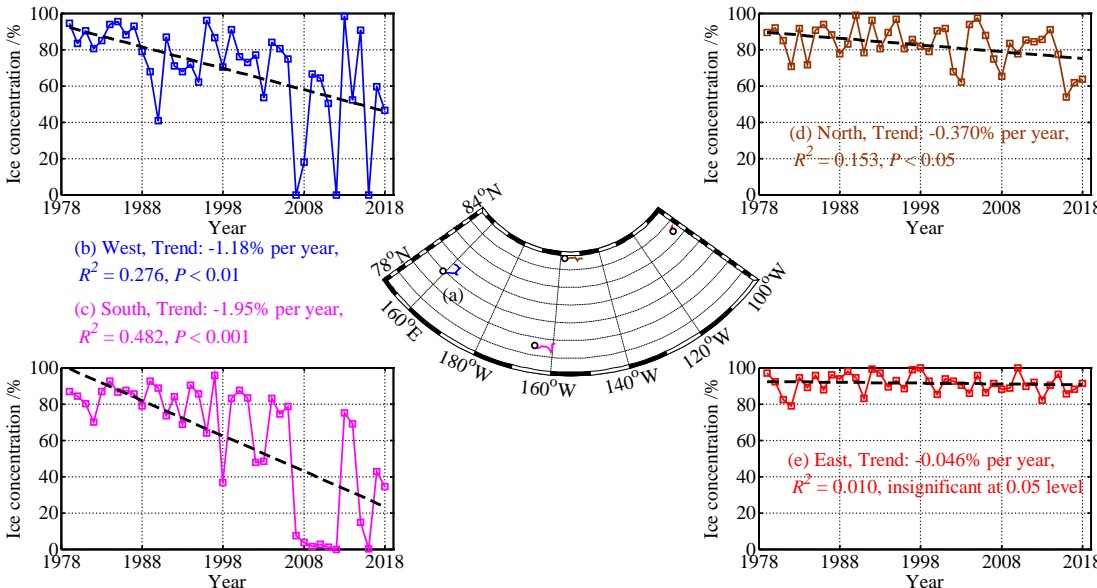


**Figure 16 (a) Drift trajectories of the westernmost, southernmost, near northernmost, and easternmost buoys**

**from 1 to 15 September 2018; the northernmost buoy has been omitted because it drifted to the north of 84.5°**

**N, where SMMR ice concentration data prior to 1987 are unavailable; trajectory of the westernmost buoy**

**was reconstructed using the NSIDC ice motion product because this buoy was deployed on 15 September**

**2018; (b–e) Long-term changes in ice concentration along buoy trajectories averaged over 1–15 September,**

**with black lines denoting linear trends.**


Table 1. Statistical relationships between IWSR and selected parameters. Significance levels are $P <$
0.001 (***), $P < 0.01$ (**), and $P < 0.05$ (*), and n.s. denotes insignificant at the 0.05 confidence level.
Numbers in parentheses indicate number of buoys used for the statistics.

| Month | vs. Lat. | vs. Lon. | vs. $W_{10m}$ | vs. $T_{2m}$ |
|---|---|---|---|---|
| 20 Aug.-30 Sep. | –0.647**(24) | –0.738***(29) | –0.542**(32) | n.s. |
| Oct. | –0.811***(24) | –0.885***(29) | –0.866***(32) | 0.657***(32) |
| Nov. | –0.777***(23) | –0.765***(28) | n.s. | 0.736***(32) |
| Dec. | –0.736***(22) | –0.829***(27) | n.s. | 0.675***(32) |
| Jan. | n.s. | –0.711**(23) | n.s. | n.s. |
| Feb. | n.s. | –0.610**(23) | n.s. | n.s. |
