# Peer review of "Seasonal changes in sea ice kinematics and deformation in the Pacific Sector of the Arctic Ocean in 2018/19"

_The Cryosphere, 2020_

## Referee Comment (RC1) · Anonymous Referee #1 · 9 Sep 2020

It is not clear from the conclusions, abstract and results where the emphasis is in this paper, with too much attention paid on the synoptic conditions over the key finding. I think the key point is that the space-time coupling for ice deformation changes over the transition from free drift to a consolidated ice pack. This point is worth reporting, as I believe it has not been shown with clarity before. However there is some points to address to make sure that this result is real.

The study also shows a gradient in response to wind forcing across the Canada Basin that might be attributed to the different ice ages. It is shown that there is increasing localization of deformation as the ice pack become more consolidated, which is echoing

work by Stern and Lindsay (2009).

I have some concerns with the methodology as presented.

1. If you just consider the amplitude of semi-diurnal peak in the velocity you are mixing measurement noise and background energy cascade (typically red noise for ice drift) with the inertial motion. How can you be sure that you are actually not aliasing the inertial power due to weather changes? Are you really sure the peak is apparent for all months? You need to consider how high above the background the inertial peak sits. In some parts of the Arctic this peaks is tidal as well as inertial. You should comment on the roll of tides in the study region.

2. Can you comment on how accurately you can estimate the area localization, delta_15%, given the sparse nature of the buoy array? Is the trend in figure 14 statistically significant?

3. Regarding the results, some are not consistent with previous studies. However there is insufficient information in the manuscript to identify if the results are reasonable based on the data. Your beta values, the spatial scaling exponent, are somewhat higher than values found in previous studies. I am referring to figure 12.

A similar decrease in beta with sampling interval, the space-time coupling, was found by Hutchings et al. 2018, who only had data for March through May. It is interesting that you find c (the gradient in log space) increases from a time the pack is in free drift to a time it is more consolidated pack. I have one suggestion to make sure your results are robust: Is there sufficient data to identify beta in only one month? I have looked at this myself and find the results to be quite messy when I split time series of buoys array deformation by month.

Incidentally there are many places in the paper where the language is implying something causes the other, such as more consolidated ice pack causes lower beta and higher c. I would suggest you consider that patterns that covary do not indicate they
cause one another, but perhaps they could be related. Consider being careful with your language throughout.

The paper could be refocused in the abstract, discussion and conclusion to focus attention on the main findings. While the synoptic situation is important and it needs to be mentioned how the ice pack responded dynamically to seasonal synoptic changes, these detail distract from the main points.

Specific points

line 21: It is not clear what "Areal localization index" is in the abstract. Perhaps use plain language here rather than jargon.

Please check for small grammatical errors. For example line 28 in the abstract "ore pronounced in the future as sea ice losses at higher rates in the". I think "as ... " should be "as sea ice losses are at higher ..."

line 35: "the Arctic Amplification". the not needed, and elsewhere.

line 43: The first sentence is hanging here, I think you need to clarify what you mean by deformation.

line 68/69: "inertial signal". You need a better description of the inertial oscillation of the ice-ocean boundary layer in response to impulses imparted by sudden changes in wind direction.

line 108, using semi-colons will help separate items in the list.

line 116: "From" should be "Of"

line 129: remove "have"

line 136: I do not understand what you are calculating over the buoys that are 1 standard deviation from mean latitude or longitude. Why choose one standard deviation? This seams arbitrary and whether there are distortion effects related to the spherical

coordinates depends on the array size, and 1 standard deviation probably changes over the time the buoy array exists.

line 156: "Because of the delayed release of NSIDC data ..". I suspect you might be able to get more recent data if you ask Mark Tshudi personally.

Regarding the inertial motion index. How do you ensure this is actually a peak and not background noise?

equations 6 and 7: I think you need to specify that beta and alpha are the scaling exponents for the mean deformation. As sea ice deformation is multifractal, the exponents vary for the different moments of the deformation distribution.

line 209, this sentence is a little clunky. I think you want to say you calculate the empirical orthogonal functions for the sea level pressure. Also, did you expand SLP earlier?

line 498-490, and line 28-29: This seams to be conjecture. The ice in this region is already mostly seasonaly any way so I think it is moot point that there will be further losses in these regions.

Finally some of the figures are overly cramped in their use of space. e.g. figure 9 almost has labels for sub panels overlapping. The month lables are hidden inside the figures and a little bit of space below the color bar would help readability. Figures 10, 15 have similar issues.

References Hutchings, J. K., Roberts, A., Geiger, C. A., & Richter-Menge, J. (2018). Corrigendum: Spatial and temporal characterization of sea-ice deformation. Journal of Glaciology, 64(244), 343-346. Stern, H. L., & Lindsay, R. W. (2009). Spatial scaling of Arctic sea ice deformation. Journal of Geophysical Research: Oceans, 114(C10).

---

## Referee Comment (RC2) · Anonymous Referee #2 · 22 Sep 2020

Based on an array of buoy measurements in the Pacific sector of the Arctic Ocean (PAO), this study examine the Arctic sea ice kinematics with great details. The association with the evolution of sea ice is identified and the linkage to the large-scale atmospheric circulation, such as Arctic Dipole, is outlined. The analysis is reasonable and the results are encouraging. Overall, this study provides very useful knowledge in reinforcing the understanding the mechanisms that regulates the Arctic sea ice kinematics. Before the recommendation for the publication, I suggest minor revision is needed.

Major comments:

[Figure]

(1) Tide is an important contributor to sea ice deformation. Thus, the discussion about the effects of tide is of interest to improve the understanding of this study.

(2) The results of this study is insightful. However, to make the results more robust, some comparisons between results of this study and those of other regions or satellite observations are encouraging.

(3) Arctic sea ice decline is in a faster track and the ecological impacts are more apparent. Therefore, it would be useful to discuss the association between sea ice deformation and Arctic sea ice decreases and related ecological process.

Minor comments:

L29, "western parts" -> "eastern parts"? L37, "enhanced Arctic Dipole (Lei et al., 2016)-> some other references may be relevant, such as:

Bi, H., Yang, Q., Liang, X., Zhang, L., Wang, Y., Liang, Y., and Huang, H., 2019, Contributions of advection and melting processes to the decline in sea ice in the Pacific sector of the Arctic Ocean. The Cryosphere, 13, 1423-1439. Ding, Q., et al., 2017, Influence of high-latitude atmospheric circulation changes on summertime Arctic seaice. Nature Climate Change, 7, 289-295.

L97 "for example" -> ", for example," L116 "From" -> "of"

Figures 9 and 10 need rearrangement to make it clearer.

---

## Author Comment (AC1) · 22 Sep 2020

Reply to reviewer 1

1 It is not clear from the conclusions, abstract and results where the emphasis is in this paper, with too much attention paid on the synoptic conditions over the key finding. I think the key point is that the space-time coupling for ice deformation changes over the transition from free drift to a consolidated ice pack. This point is worth reporting, as I believe it has not been shown with clarity before. However there are some points to address to make sure that this result is real. We will rewrite the sectors of conclusions, abstract and results, and make them more focusing. We will highlight the space-time

coupling for kinematics and ice deformation changes over the transition from free drift to a consolidated ice pack.

2 The study also shows a gradient in response to wind forcing across the Canada Basin that might be attributed to the different ice ages. It is shown that there is increasing localization of deformation as the ice pack become more consolidated, which is echoing work by Stern and Lindsay (2009). We will further compare our results with theirs (Stern and Lindsay, 2009), and highlight the spatial gradient of ice kinematics and ice deformation in response to wind forcing in the conclusions and abstract.

3 If you just consider the amplitude of semi-diurnal peak in the velocity you are mixing measurement noise and background energy cascade (typically red noise for ice drift) with the inertial motion. How can you be sure that you are actually not aliasing the inertial power due to weather changes? Are you really sure the peak is apparent for all months? You need to consider how high above the background the inertial peak sits. In some parts of the Arctic this peaks is tidal as well as inertial. You should comment on the roll of tides in the study region. Inertial oscillations (in the northern hemisphere) are clockwise oscillations, in contrast to tidal oscillation, which can rotate clockwise or counter-clockwise. Amplitudes shown in Figure 9 are that at the local negative inertial frequency (about -2.01 $\sim$ -1.94) after Fourier transformation of monthly time series of normalized ice velocity. At this frequency, there are also some energy caused by tidal forcing and high-frequent parts of wind and current forcing. In the revision, we will also show the amplitudes at the positive tidal frequency (+2), which includes the energy from tidal forcing and background noise of high-frequent parts of wind and current forcing. From the amplitudes at the positive tidal frequency, we cannot identify the obvious seasonal and spatial variations because all the buoys were deployed over the deep waters and the tidal forcing is relatively weak. In addition, both tidal forcing and high-frequent parts of wind and current forcing are not expected to have seasonal changes. Thus, we will further explain the spatiotemporal change patterns shown in Figure 9 are majorly attributed to the changes caused by inertial oscillations.

3. Can you comment on how accurately you can estimate the area localization, delta_15%, given the sparse nature of the buoy array? Is the trend in figure 14 statistically significant? To estimate the area fraction of top 15% ice deformation, we use the data obtained from the relatively dense buoy array deployed in the north of Pacific sector of Arctic Ocean, but not from all buoys. We will test the reliability for using the area fraction of 15% to estimate the area localization through using various fractions. The trend in Fig. 14 is statically significant at 0.01 level.

4 Regarding the results, some are not consistent with previous studies. However there is insufficient information in the manuscript to identify if the results are reasonable based on the data. Your beta values, the spatial scaling exponent, are somewhat higher than values found in previous studies. I am referring to figure 12. The spatial scaling exponent is strongly dependent on the ice cohesiveness and temporal sampling rate. In the Fig. 12, the results include the results obtained from September and at the 1-h temporal resolution, thus including some relatively large value. The value obtained in Jan-Feb. at the 3-h temporal resolution shown in Fig. 2 was 0.35–0.37, which was slightly smaller than that obtained in Beaufort Sea during March-May 2007 (0.40), and comparable with that obtained in the Central Artic Ocean during May 2007 (Itkin et al., 2017). As our known, the Arctic sea ice in the Pacific Sector may reach to its annual maximum thickness in May or early Jun. (e.g., Perovich et al., 2003). Thus, the strongest ice cohesiveness would occur during the latter winter and early spring, but not in the mid-winter when the air temperature is coldest but the ice thickness still doesn't reach the annual maximum. In the revision, we can enhance the comparison with the results from other studies.

Perovich, D. K., T. C. Grenfell, J. A. Richter-Menge, B. Light, W. B. Tucker III, and H. Eicken, Thin and thinner: Sea ice mass balance measurements during SHEBA, J. Geophys. Res., 108(C3), 8050, doi:10.1029/2001JC001079, 2003.

5 A similar decrease in beta with sampling interval, the space-time coupling, was found by Hutchings et al. 2018, who only had data for March through May. It is interesting

that you find c (the gradient in log space) increases from a time the pack is in free drift to a time it is more consolidated pack. I have one suggestion to make sure your results are robust: Is there sufficient data to identify beta in only one month? I have looked at this myself and find the results to be quite messy when I split time series of buoys array deformation by month. As mentioned above, we will highlight the new findings for the space-time coupling of ice deformation. By combining the findings obtained from Hutchings et al. (2018), we will add some discussions on the annual circle of the space-time coupling regime of ice deformation. To estimate the beta, we use the strain rate obtained from all triangles consisting of any three buoys, which can guarantee the magnitude of statistical samples. This method has been used by Itkin et al. (2017), who also estimate the beta using the data obtained from one month. To test if our results are robust, we will further estimate the seasonal beta, i.e., those obtained in autumn (September-November) and winter (December-February).

6 Incidentally there are many places in the paper where the language is implying something causes the other, such as more consolidated ice pack causes lower beta and higher c. I would suggest you consider that patterns that covary do not indicate they cause one another, but perhaps they could be related. Consider being careful you're your language throughout. Thanks for the suggestions. We will check the language through the manuscript and make sure that the expression is rigorous and clear.

7 The paper could be refocused in the abstract, discussion and conclusion to focus attention on the main findings. While the synoptic situation is important and it needs to be mentioned how the ice pack responded dynamically to seasonal synoptic changes, these details distract from the main points. Thanks for the suggestions. In the revision, we will focus on the seasonal changes in the space-time coupling of ice deformation.

8 line 21: It is not clear what "Areal localization index" is in the abstract. Perhaps use plain language here rather than jargon. We will use the plain language in the abstract.

9 Please check for small grammatical errors. For example line 28 in the abstract "ore

pronounced in the future as sea ice losses at higher rates in the". I think "as ... " should be "as sea ice losses are at higher ..." We will check the grammatical errors through the manuscript.

10 line 43: The first sentence is hanging here, I think you need to clarify what you mean by deformation. We will correct this mistake in expression.

11 line 68/69: "inertial signal". You need a better description of the inertial oscillation of the ice-ocean boundary layer in response to impulses imparted by sudden changes in wind direction. We will add the discussions on the inertial oscillation of the ice-ocean boundary layer in response to impulses imparted by sudden changes in wind direction.

12 line 108, using semi-colons will help separate items in the list. line 116: "From" should be "Of" line 129: remove "have" We will correct these mistakes in expression.

13 line 136: I do not understand what you are calculating over the buoys that are 1 standard deviation from mean latitude or longitude. Why choose one standard deviation? This seams arbitrary and whether there are distortion effects related to the spherical coordinates depends on the array size, and 1 standard deviation probably changes over the time the buoy array exists. We will give detail on the changes in the geographical distance according our use of 1 standard deviation of latitude and longitude.

14 line 156: "Because of the delayed release of NSIDC data ..". I suspect you might be able to get more recent data if you ask Mark Tshudi personally. It's just supporting data, but we will try to discuss with Mark Tshudi.

15 Regarding the inertial motion index. How do you ensure this is actually a peak and not background noise? In fact, the peak value of inertial oscillation will be affected by the high frequency variations of wind or current, but the influence is very small. We select the peak value manually in the range of 0.5h near the inertial period. In the reversion, we will further explain the method.

16 equations 6 and 7: I think you need to specify that beta and alpha are the scaling exponents for the mean deformation. As sea ice deformation is multifractal, the exponents vary for the different moments of the deformation distribution. We will specify that beta and alpha are the scaling exponents for the mean deformation.

17 line 209, this sentence is a little clunky. I think you want to say you calculate the empirical orthogonal functions for the sea level pressure. Also, did you expand SLP earlier? Yes, we want to say we calculate the empirical orthogonal functions for the sea level pressure. We will make the expression clearer. We have expanded SLP already in Line 146.

18 line 498-490, and line 28-29: This seams to be conjecture. The ice in this region is already mostly seasonally any way so I think it is moot point that there will be further losses in these regions. Yes, the ice in these regions is already mostly seasonally. However, the further lengthened ice melting period, even the length of free-ice waters occupation, will shorten the growth season of sea ice and reduce the ice thickness, thus enhancing the response of sea ice kinematics and dynamic deformation to atmospheric forcing. We will add some discussions on this feedback regiem.

19 Finally some of the figures are overly cramped in their use of space. e.g. figure 9 almost has labels for sub panels overlapping. The month lables are hidden inside the figures and a little bit of space below the color bar would help readability. Figures 10, 15 have similar issues. We will improve these figures.

---

## Author Comment (AC2) · 27 Sep 2020

Reply to reviewer 2

1 Tide is an important contributor to sea ice deformation. Thus, the discussion about the effects of tide is of interest to improve the understanding of this study. –Yes, tide is an important contributor to sea ice deformation, especially over the shallow waters. However, using the buoys data, it is hard to identify the effect of tide forcing on ice deformation. Firstly, we will add some qualitative discussions on the effect of tide forcing on ice deformation, which is relatively weak in the deep basin, where the buoy array were deployed; secondly, we will add some spectrum analysis to identify the

influence of tide forcing on the quasi semidiurnal oscillation of ice motion.

2 The results of this study are insightful. However, to make the results more robust, some comparisons between results of this study and those of other regions or satellite observations are encouraging. –Thanks for the suggestions. To enhance the representativeness of our results and give some basin-scale implications for the ice dynamics, we will add some comparisons with results obtained from other regions or the close region in other years, as well as that obtained from the estimations based on satellite observations.

3 Arctic sea ice decline is in a faster track and the ecological impacts are more apparent. Therefore, it would be useful to discuss the association between sea ice deformation and Arctic sea ice decreases and related ecological process. –We will add some discussions on the implications of enhanced ice deformation on Arctic ice loss and some ice-associated ecological processes.

4 L29, "western parts" -> "eastern parts"? L97 "for example" -> ", for example," L116 "From" -> "of" –We will correct these linguistic errors, and check the language through the manuscript again.

5 L37, "enhanced Arctic Dipole (Lei et al., 2016)-> some other references may be relevant, such as: Bi, H., Yang, Q., Liang, X., Zhang, L., Wang, Y., Liang, Y., and Huang, H., 2019, Contributionsof advection and melting processes to the decline in sea ice in the Pacific sector of the Arctic Ocean. The Cryosphere, 13, 1423-1439. Ding, Q., et al., 2017, Influence of high-latitude atmospheric circulation changes on summertime Arctic sea ice. Nature Climate Change, 7, 289-295. –We will cited these two references and enhance the discussions on the influence of atmospheric circulation on ice motion.

---

## Author Comment (AC3) · 23 Oct 2020

Q:In some parts of the Arctic this peaks is tidal as well as inertial. You should comment on the roll of tides in the study region. Re: Inertial oscillations (in the northern hemisphere) are clockwise oscillations, in contrast to tidal oscillation, which can rotate clockwise or counter-clockwise. Amplitudes shown in Figure 9 are that at the local negative inertial frequency (about -2.01 $\sim$ -1.94) after Fourier transformation of monthly time series of normalized ice velocity. At this frequency, there are also some energy caused by tidal forcing and high-frequent parts of wind and current forcing. In the revision, the amplitudes at the positive tidal frequency (+2) includes the energy from tidal

forcing and background noise of high-frequent parts of wind and current forcing. At positive tidal frequency (+2), the signal is very weak, which is obviously lower than the value of negative phase in any season, and there is no seasonal change. Therefore, we believe that the obvious regional and spatial changes of negative signal are caused by inertial forcing.

Q: Can you comment on how accurately you can estimate the area localization, delta_15%, given the sparse nature of the buoy array? Re: When we use the area fraction of 10% and 20% to estimate the area localization, the results are still robust, and similar with that obtained using 15%.

Q: I have one suggestion to make sure your results are robust: Is there sufficient data to identify beta in only one month? Re: We also calculate use the two-month temporal windows, i.e., Sep-Oct, Nov-Dec, Jan-Feb. Although the values show some differences, the seasonal change pattern for the beta is similar, thus we believe that the results are robust. We also comment that the values of calculated beta are sensitive to changes in temporal window.

---

## Author Response (AR1)

**Dear Editor,**

We would like to submit our revised manuscript entitled "**Seasonal changes in sea ice kinematics and deformation in the Pacific Sector of the Arctic Ocean in 2018/19**" [Paper # tc-2020-211] , which is submitted for possible publication in the the Cryosphere. According to the comments from anonymous reviewers and you, we made a major revision of the manuscript by carrying out the following tasks:

1) We restructured the manuscript, especially for the sectors of Introduction, Discussion, and Conclusions, to make it more compact and focusing.
2) In terms of methodology, we improved it to make the expression clearer and added some sensitivity calculation. In particular, we have added an explanation of how to distinguish contributions from inertial and tidal oscillations to sea ice kinematics.
3) We enhanced the comparison with the results from other studies.
4) We checked the language through the manuscript, revised some figures, and made the expressions clearer.
5) We added some discussions on the implications of enhanced ice deformation on the biological processes.

Please find the following files in our submission package:
1) The manuscripts with tracked changes,
and 2) Response letter.

Thank you for your time.
Sincerely,
Ruibo Lei, other co-authors

It is not clear from the conclusions, abstract and results where the emphasis is in this paper, with too much attention paid on the synoptic conditions over the key finding. I think the key point is that the space-time coupling for ice deformation changes over the transition from free drift to a consolidated ice pack. This point is worth reporting, as I believe it has not been shown with clarity before. However there are some points to address to make sure that this result is real.

We restructured the sectors of abstract, introductions, and conclusions, and made them more focusing on the seasonal and spatial changes in sea ice kinematics and deformation, as well as their coupling. In terms of methodology, we improved it to make the expression clearer and added some sensitivity calculation. For example, we have added an explanation of how to distinguish contributions from inertial and tidal oscillations to sea ice motion (Lines 207-230, the line number refer to the revised manuscript with track changes); to test the sensitivity of the estimation of β to the number of samples, we also calculate the length scaling law using seasonal temporal window (Lines 494-498); to estimate the localization of ice deformation, we further use other thresholds to calculate the fractional areas accommodating the largest the ice deformation (Lines 529-534).

The study also shows a gradient in response to wind forcing across the Canada Basin that might be attributed to the different ice ages. It is shown that there is increasing localization of deformation as the ice pack become more consolidated, which is echoing work by Stern and Lindsay (2009).

We further compared our results with Stern and Lindsay (2009) (Lines 111-113; 492-494; 527), and highlight the spatial gradient of ice kinematics and ice deformation in response to wind forcing.

If you just consider the amplitude of semi-diurnal peak in the velocity you are mixing measurement noise and background energy cascade (typically red noise for ice drift) with the inertial motion. How can you be sure that you are actually not aliasing the inertial power due to weather changes? Are you really sure the peak is apparent for all months? You need to consider how high above the background the inertial peak sits. In some parts of the Arctic this peaks is tidal as well as inertial. You should comment on the roll of tides in the study region.

Inertial oscillations are clockwise oscillations in the northern hemisphere, in contrast to tidal oscillation, which can rotate clockwise or counter-clockwise (Gimbert et al., 2012).

Amplitudes shown in the original Figure 9 are that at the local maximum of negative inertial frequency (about -2.01 ~ -1.94 cycles per day) after complex Fourier transformation of monthly time series of normalized ice velocity. At this frequency, there is energy caused by inertial and tidal forcing and high-frequent components of wind and current forcing. In the revision, we also identify and show the amplitudes at the positive tidal frequency (+2 cycles per day) in the current Figure 9, which includes the energy from tidal forcing and background noise of high-frequent components of wind and current forcing. From the amplitudes at the positive tidal frequency, we cannot identify the obvious seasonal and spatial variations because all the buoys were deployed over the deep waters and the tidal forcing is relatively weak. In addition, both tidal forcing and high-frequent parts of wind and current forcing are not expected to have seasonal changes. Thus, the spatiotemporal change patterns of the amplitudes at negative inertial frequency are majorly attributed to the changes caused by inertial oscillations. We have added an explanation of how to distinguish contributions from inertial and tidal oscillations to sea ice motion (Lines 207-230; Line 371-391).

3. Can you comment on how accurately you can estimate the area localization, delta_15%, given the sparse nature of the buoy array? Is the trend in figure 14 statistically significant?

To estimate the area fraction of 15% largest ice deformation, we use the data obtained from the relatively dense buoy array deployed in the north of Pacific sector of Arctic Ocean, but not from all buoys (Line 260). We test the reliability for using the area fraction of 15% to estimate the area localization through using various fractions. (Lines 529-534)

The trend in Fig. 14 is statically significant at 0.001 level (Line 513).

Regarding the results, some are not consistent with previous studies. However there is insufficient information in the manuscript to identify if the results are reasonable based on the data. Your beta values, the spatial scaling exponent, are somewhat higher than values found in previous studies. I am referring to figure 12.

The spatial scaling exponent is strongly dependent on the ice cohesiveness and temporal sampling rate. In the Fig. 12, the results include the results obtained from September and at the 1-h temporal resolution, thus including some relatively large values. The value obtained in Jan-Feb. with the 3-h temporal resolution was 0.42–0.44, which was comparable with that obtained in Beaufort Sea during March-May 2007 (0.40) (Itkin et al., 2017). As our known, Arctic sea ice in the Pacific Sector may reach to its annual maximum thickness in May or early Jun. (e.g., Perovich et al., 2003). The strongest ice cohesiveness would occur during the latter winter or early spring. This is because, in the mid-winter, the ice thickness still doesn't reach the annual maximum although the air temperature is coldest. In the revision, we enhanced the comparison with the results from other studies. (Lines 486-494)

A similar decrease in beta with sampling interval, the space-time coupling, was found by Hutchings et al. 2018, who only had data for March through May. It is interesting that you find c (the gradient in log space) increases from a time the pack is in free drift to a time it is more consolidated pack. I have one suggestion to make sure your results are robust: Is there sufficient data to identify beta in only one month? I have looked at this myself and find the results to be quite messy when I split time series of buoys array deformation by month.

As mentioned above, we highlighted the new findings for the space-time coupling of ice deformation.

To estimate the beta, we use the strain rate obtained from all triangles consisting of any three buoys, which can guarantee the magnitude of statistical samples. This method is a little different from that given by Hutchings et al. (2018), who estimated the deformation rate using the fixed buoy-triangle groups, and has been used by Itkin et al. (2017), who also estimate the beta using the data obtained from one month. (Line 218)

To test if our results are robust, we also estimate the seasonal beta, i.e., those obtained in autumn (September-November) and winter (December-February). (Lines 495-498)

Incidentally there are many places in the paper where the language is implying something causes the other, such as more consolidated ice pack causes lower beta and higher c. I would suggest you consider that patterns that covary do not indicate they cause one another, but perhaps they could be related. Consider being careful you're your language throughout.

Thanks for the suggestions. We checked the language through the manuscript and made sure that the expression is rigorous and clear.

The paper could be refocused in the abstract, discussion and conclusion to focus attention on the main findings. While the synoptic situation is important and it needs to be mentioned how the ice pack responded dynamically to seasonal synoptic changes, these details distract from the main points.

Thanks for the suggestions. In the revision, we focus on the seasonal changes in the ice deformation and its space-time coupling. We restructured the sectors of abstract, introductions, and conclusions, and made them more compact and focusing.

line 21: It is not clear what "Areal localization index" is in the abstract. Perhaps use plain language here rather than jargon.

We used the plain language in the abstract. (Lines 22-23)

Please check for small grammatical errors. For example line 28 in the abstract "ore pronounced in the future as sea ice losses at higher rates in the". I think "as ... " should be "as sea ice losses are at higher ..."

We checked the grammatical errors through the manuscript.

line 43: The first sentence is hanging here, I think you need to clarify what you mean by deformation.

We corrected this mistake in expression (Lines 44-45).

line 68/69: "inertial signal". You need a better description of the inertial oscillation of the ice-ocean boundary layer in response to impulses imparted by sudden changes in wind direction.

We added the discussions on the inertial oscillation of the ice-ocean boundary layer in response to impulses imparted by sudden changes in wind direction. (Lines 81-83;

)

line 108, using semi-colons will help separate items in the list. line 116: "From" should be "Of" line 129: remove "have"

We corrected these mistakes in expression. (Lines 130-137; Line 141)

line 136: I do not understand what you are calculating over the buoys that are 1 standard deviation from mean latitude or longitude. Why choose one standard deviation? This seams arbitrary and whether there are distortion effects related to the spherical coordinates depends on the array size, and 1 standard deviation probably changes over the time the buoy array exists.

We gave details on the changes in the geographical distance according to our use of 1 standard deviation of latitude and longitude; and added some discussions on the reliability of this treatment method. (Lines 164-171)

line 156: "Because of the delayed release of NSIDC data ..". I suspect you might be able to get more recent data if you ask Mark Tshudi personally.

We are using the latest version of the data, which is updated by December 2018 (Version 4; Tschudi et al., 2019 and 2020). Because this is just the supporting data, we consider that the lack of some comparative data will not have a significant impact on our results.

Regarding the inertial motion index. How do you ensure this is actually a peak and not background noise?

In fact, the peak value of inertial oscillation might be affected by the high frequency variations of wind or current, but the influence is very small. We selected the peak value manually in the range of ±0.03 cycles per days from the targeted frequency. In the reversion, we further explained the method. When the inertial oscillation is very weak (with ~15% cases), we use the amplitudes at $-f_0$ as the IMI, which actually is background noise, but also can indicate a state with an almost negligible inertial oscillation occurred only in winter. Therefore, these conditions will not affect our identification of the seasonal variation of the contribution of the inertial oscillation on ice motion. (Lines 207-230)

equations 6 and 7: I think you need to specify that beta and alpha are the scaling exponents for the mean deformation. As sea ice deformation is multifractal, the exponents vary for the different moments of the deformation distribution.

We specified that beta and alpha are the scaling exponents for the mean deformation. (Lines 323-325)

line 209, this sentence is a little clunky. I think you want to say you calculate the empirical orthogonal functions for the sea level pressure. Also, did you expand SLP earlier?

Yes, we calculate the empirical orthogonal functions for the sea level pressure. We made the expression clearer (Line 345-346). We have expanded SLP already in Line 235.

line 498-490, and line 28-29: This seems to be conjecture. The ice in this region is already mostly seasonally any way so I think it is moot point that there will be further losses in these regions.
Yes, the ice in these regions is already mostly seasonally in the west and south parts for PAO. However, the further lengthened ice melt season, and the increased length of free-ice waters occupation, will shorten the growth season of sea ice and reduce the ice thickness, thus enhancing the response of sea ice kinematics and dynamic deformation to atmospheric forcing. We added some discussions on this feedback regime. (Lines 755-769)

Finally some of the figures are overly cramped in their use of space. e.g. figure 9 almost has labels for sub panels overlapping. The month lables are hidden inside the figures and a little bit of space below the color bar would help readability. Figures 10, 15 have similar issues.
We improved these figures.

**Reply to reviewer 2**
Tide is an important contributor to sea ice deformation. Thus, the discussion about the effects of tide is of interest to improve the understanding of this study.
Yes, tide is an important contributor to sea ice deformation, especially over the shallow waters. However, using the buoys data, it is hard to identify the effect of tide forcing on ice deformation directly.
In the revised manuscript, we added some quantitative discussions on the effect of tidal forcing on ice motion (Lines 295-298; 534-537). Compared with the inertial oscillation, the contribution of tidal oscillation is relatively weak, which gives a negligible contribution to the seasonal and spatial changes on the quasi-semidiurnal oscillation of ice motion. From this analysis, we infer that the influence of tidal forcing on ice deformation also is negligible regardless of seasons (Lines 534-537).

The results of this study are insightful. However, to make the results more robust, some comparisons between results of this study and those of other regions or satellite observations are encouraging.
Thanks for the suggestions. To enhance the representativeness of our results and give some basin-scale implications for the ice dynamics, we added some comparisons with results obtained from close regions, as well as that obtained from the estimations based on satellite observations. (Lines 688-696; 714-721; 734-738)

Arctic sea ice decline is in a faster track and the ecological impacts are more apparent. Therefore, it would be useful to discuss the association between sea ice deformation and Arctic sea ice decreases and related ecological process.

We added some discussions on the implications of enhanced ice deformation on some ice-associated ecological processes. (Lines 939-951); and the association between Arctic sea ice deformation and sea ice decrease (Lines 755-769).

L29, "western parts" -> "eastern parts"? L97 "for example" -> ", for example," L116 "From" -> "of"

We corrected these linguistic errors, and checked the language through the manuscript again.

L37, "enhanced Arctic Dipole (Lei et al., 2016)-> some other references may be relevant, such as:

Bi, H., Yang, Q., Liang, X., Zhang, L., Wang, Y., Liang, Y., and Huang, H., 2019, Contributionsof advection and melting processes to the decline in sea ice in the Pacific sector of the Arctic Ocean. The Cryosphere, 13, 1423-1439.

Ding, Q., et al., 2017, Influence of high-latitude atmospheric circulation changes on summertime Arctic sea ice. Nature Climate Change, 7, 289-295.

We cited these two references and enhanced the discussions on the influence of atmospheric circulation on ice motion (Lines 87-91; 368).

Figures 9 and 10 need rearrangement to make it clearer.

We improved these figures.

**Seasonal changes in sea ice kinematics and deformation in the Pacific Sector of the Arctic Ocean in 2018/19**

Ruibo Lei[1], Mario Hoppmann[2], Bin Cheng[3], Guangyu Zuo[1,4], Dawei Gui[1,5], Qiongqiong Cai[6], H. Jakob Belter[2], Wangxiao Yang[4]

[1] Key Laboratory for Polar Science of the MNR, Polar Research Institute of China, Shanghai, China.

[2] Alfred-Wegener-Institut Helmholtz-Zentrum für Polar- und Meeresforschung, Bremerhaven, Germany.

[3] Finnish Meteorological Institute, Helsinki, Finland.

[4] College of Electrical and Power Engineering, Taiyuan University of Technology, Taiyuan, China.

[5] Chinese Antarctic Center of Surveying and Mapping, Wuhan University, Wuhan, China.

[6] National Marine Environmental Forecasting Center of the MNR, Beijing, China.

*Correspondence to*: Ruibo Lei (leiruibo@pric.org.cn)

**Abstract.** Arctic sea ice kinematics and deformation play significant roles in heat and momentum exchange between atmosphere and ocean, and they have a profound impact on biological processes and biogeochemical cycles. However, mechanisms regulating their changes at seasonal scales remain poorly understood. Using position data of 32 buoys in the Pacific sector of the Arctic Ocean (PAO), we characterized spatiotemporal variations in ice kinematics and deformation for autumn–winter 2018/19 over the transition from melting ice to a near consolidated ice pack. In autumn, the response of sea ice drift  to wind forcing and its inertial oscillation were stronger in the southern and western than in the northern and eastern parts of the PAO. These spatial heterogeneities decreased gradually from autumn to winter, in line with the increases in  ice concentration and thickness. Correspondingly, ice deformation becomes much more localized as the sea ice mechanical strength increases, with the area proportion occupied by the strongest ice deformation  decreasing by about 50 % from autumn to winter During the freezing season, ice deformation rate in the northern part of the PAO was about 2.5 times that in the western part probably related to the higher spatial heterogeneity of oceanic and atmospheric forcing in the north. North–south and east–west gradients in sea ice kinematics and deformation of the PAO, as observed in this study, are likely to become more pronounced in the future as a result of a longer melt season, especially in the western and southern  parts.

**1 Introduction**

The Pacific sector of Arctic Ocean (PAO) includes the Beaufort, Chukchi, and East Siberian Seas, as well as the Canadian and Makarov Basins. Among all the sectors of the Arctic Ocean, decreases in both summer sea ice (Comiso et al., 2017) and multi-year sea ice (MYI) (Serreze and Meier, 2018) are the largest in the PAO in recent decades, and are most likely linked to enhanced ice–albedo feedback (Steele and Dickinson, 2016), increased Pacific water inflow (Woodgate et al., 2012), and enhanced Arctic

Dipole (Lei et al., 2016). In the PAO, MYI is mainly distributed north of the Canadian Arctic

Archipelago (Lindell and Long, 2016), suggesting a strong east–west gradient in sea ice thickness and strength. In summer, the marginal ice zone (MIZ), defined as the area where sea ice concentration is less than 80 %, can reach as far north as 80° N (Strong and Rigor, 2013), thus the south–north gradient in ice conditions in the PAO is expected to be greater than that in other sectors of the Arctic Ocean.

Sea ice deformation results from divergence, convergence, and shear , which can enhance redistribution of ice thickness through the formation of leads and ridges (Hutchings and Hibler, 2008;

Itkin et al., 2018). Loss of MYI and decreased thickness weakens the Arctic sea ice cover, increases floe mobility (Spreen et al., 2011), and promotes ice deformation (Kwok, 2006)

. Leads forming between ice floes increase heat transfer  from the  ocean to the atmosphere, a process that is particularly important in winter because of the large temperature gradient (Alam and Curry, 1998)

. In summer, cracks, leads or polynyas within c the pack ice serve as windows that expose the ocean to more sunlight, which significantly alters biological processes and biogeochemical cycles such as promoting under-ice haptophyte algae blooms (Assmy et al., 2017). Under converging conditions, ice blocks are packed randomly during the formation of sea ice pressure ridges, creating water-filled voids that act as thermal buffers for subsequent ice growth (Salganik et al., 2020). The high porosity of pressure ridges createsresults in an abundance of nutrients for ice algae communities. As a result, pressure ridges can become biological hotspots (Fernández-Méndez et al., 2018). Thus, characterizations of sea ice deformation are not only relevant for a better understanding of ice dynamics and their roles in Arctic climate current changes in Arctic climate system, but especially and also of ice-associated ecosystems.

In the PAO, the generally anticyclonic Beaufort Gyre (BG) generates sea ice motion that is clockwise on average. The boundary and strength of the BG are mainly regulated by the Beaufort High (BH)

(Proshutinsky et al., 2009; Lei et al., 2019). An aAnomalously low BH can result in a reversal of wind and ice motion in the PAO that is normally anticyclonic (Moore et al., 2018). Under a positive Arctic

Dipole Anomaly (DA), more sea ice from the PAO is transported to the Atlantic sector of the Arctic

Ocean (AAO), i.e., promoting ice advection from the BG system to the Transpolar Drift Stream (TDS)

(Wang et al., 2009). In summer, such a regime would stimulate the ice–albedo feedback and accelerate sea ice retreat (Lei et al., 2016). The loss of summer sea ice in the PAO during the recent four decades can be explained using the increased ice advection from the PAO to the AAO by 9.6% (Bi et al., 2019).

In the zonal direction, the enhanced anticyclonic circulation in the PAO can result in more ice advection from the Beaufort and Chukchi seas to the East Siberian Sea (Ding et al., 2017). The rResponse of sea ice advection in this region to interannual variation of atmospheric circulation patterns has been studied extensively (e.g., Vihma et al., 2012), but investigations on a seasonal scale are relatively scarce.

From a dynamical perspective, sea ice consolidation has been quantified related tousing the strength of the inertial signal of sea ice motion (Gimbert et al., 2012), Ice–Wind Speed Ratio (IWSR) (Haller et al.,

2014), localization, intermittence and space–time coupling of sea ice deformation (Marsan et al., 2004), as well as to the response of ice deformation to wind forcing (Haller et al., 2014). The inertial oscillation is caused by the earth's rotation and is stimulated by sudden changes in external forces, majorly due to enhanced wind stress on the ice-ocean mixing layer caused by storms/cyclones or moving fronts of extreme weather eventsbecause of storms or moving fronts (e.g., Lammert et al., 2009; Gimbert et al.,

2012). It wouldis be weakened due to surface friction and internal ice stresses. The localization and intermittence of sea ice deformation indicate the degree of constraint for the its spatial range and temporal duration of sea ice deformation (Rampal et al., 2008). Space-time coupling demonstrates the temporal or spatial dependence for the spatial or temporal scaling laws of ice deformation, which can indicate the brittle behaviour of sea ice deformation (Rampal et al., 2008; Marsan and Weiss, 2010).

The inertial oscillations of Arctic sea ice motion (Gimbert et al., 2012) and the IWSR (Spreen et al., 2011)

have been increasingly associated with reduced sea ice thickness and  concentration.  However,  the spatial variability of  sea ice consolidation, kinematics and deformation on  seasonal scales in the PAO, where sea ice condition has strong spatial heterogeneity as mentioned above, remain unclear.

The application of drifting buoys to determine the properties and seasonal cycle of the atmosphere, ocean and sea ice on a basin scale and year-round has been an emerging field in polar research in recent years. For example, drifting buoys are a good tool to track relative ice motion. However, because of the usually limited number of such buoys deployed in any given region and season due to cost and logistical limitations, it has so far been difficult to accurately distinguish spatial variability and temporal change in sea ice kinematics and deformation from existing buoy data in the PAO. During spring 2003,  the deformation of a single lead in the Beaufort Sea  was  investigated using Global

Positioning System (GPS) receivers (Hutchings and Hibler, 2008). The ice deformation and its length scaling law in the south of the PAO during March–May have been estimated before by Hutchings et al.

[revised manuscript text omitted]

Also, some buoys  ceased operation by March 2019. Two-thirds of the buoys (22) continued to send data until or beyond the end of the study period. The trajectories of the buoys during the study period covered the region of 76° N–87° N and 155° E–110°W, which we define here as our study region.

**2.2 Analysis of sea ice kinematic characteristics**

All buoys  had a sampling interval of either 0.5 or 1 h. Prior to the calculation of ice drift velocity, position data measured by the buoys were interpolated to a regular interval ($\tau$) of 1 h. To quantify meridional (zonal) variabilities of ice kinematic properties, we used data from buoys that were within one standard deviation of the average longitude (latitude), which helps to minimize influence of zonal (meridional) difference on meridional (zonal) variabilities. This constraint leads to a meridional extent ranging from 350 to 402 km when the zonal variabilities of ice kinematics were assessed and a zonal extent ranging from 195 to 285 km for the assessment of meridional variabilities. Their seasonal changes can be considered as moderate (<40%) although the divergence of the buoys occurred at all times. If we use  half the standard deviations to constrain the calculation range, there would be no essential change in the identified meridional/zonal dependencies of ice kinematics from those obtained using one standard deviation. Thus, we consider our evaluation method as robust. Meridional variabilities are related to the transition from the

MIZ to the PIZ, while zonal variabilities  indicate the change between the region north of the

Canadian Arctic Archipelago, where MYI coverage is usually large (Lindell and Long, 2016) and the

Makarov Basin, which is mainly covered by seasonal ice (Serreze and Meier, 2018).

Two parameters were used to characterize sea ice kinematics. First, the IWSR was used to investigate the response of sea ice motion to wind forcing. Impacts of resampling data at intervals between 1 and

48 h, meridional and zonal spatial variabilities, intensity of wind forcing, near-surface air temperature, and ice concentration on the IWSR were assessed. These parameters are either related to spatiotemporal changes in atmospheric and sea ice conditions, or to the frequency characteristics of ice and wind speeds. The data used to characterize atmospheric forcing, including Sea Level air

Pressure (SLP), near-surface air temperature at 2 m ($T_{2m}$) and wind velocity at 10 m ($W_{10m}$), were obtained from the ECMWF ERA-Interim reanalysis (Dee et al., 2011). Sea ice concentration was obtained from the Advanced Microwave Scanning Radiometer 2 (AMSR2) (Spreen et al., 2008). To identify the state of atmospheric forcing and ice conditions relative to the climatology, we also calculated anomalies of SLP, $T_{2m}$, $W_{10m}$, ice concentration, and ice drift speed relative to the 1979–

2018 averages. To estimate ice concentration anomalies, we used ice concentration data from the

Nimbus-7 Scanning Multichannel Microwave Radiometer (SMMR) and its successors (SSM/I and

SSMIS) (Fetterer et al., 2017) because they cover a longer period than AMSR2 data. We used the daily product of sea ice motion (Tschudi et al., 2019 and 2020Fowler et al., 2013) provided by the

National Snow and Ice Data Center (NSIDC) to estimate ice drift speed anomalies. Because of the delayed release of NSIDC data, ice drift speed anomalies were only estimated for August–December

2018.

Second, the inertial motion index (IMI) was used to quantify the inertial component of ice motion. To obtain the IMI, we applied a Fast Fourier Transformation to normalized hourly ice velocities.

Normalized ice velocities were calculated by scaling velocity values to monthly averages, allowing seasonal change to be assessed independently of magnitudes of ice velocities. The frequency of the inertial oscillation varies with latitude as follows:

$$f_0 = 2\Omega \sin \theta \tag{1}$$

where $f_0$ is inertial frequency, $\Omega$ is Earth rotation rate, and $\theta$ is latitude. The $f_0$ ranges from 2.01 to 1.94

cycles day$^{-1}$ between 90° N and 75° N. Rotary spectra calculated from sea ice velocity using complex

Fourier analysis were used to identify signals of inertial and tidal origin, both of which have a frequency of ~ 2 cycles day$^{-1}$ in the Arctic Ocean (Gimbert et al., 2012). According to Gimbert et al.

(2012), the complex Fourier transformation $\widehat{U}(\omega)$ is defined as:

$$\widehat{U}(\omega) = \frac{1}{N} \sum_{t=t_0}^{t_{end}-\Delta t} e^{-i\omega t} \left( u_x + i u_y \right), \tag{2}$$

where $N$ and $\Delta t$ are the number and temporal interval of velocity samples, $t_0$ and $t_{end}$ are the start and end times of the temporal window, $u_x$ and $u_y$ are zonal and meridional ice speeds at $t+0.5\Delta t$ on an orthogonal geographical grid, and $\omega$ is angular frequency. The IMI was defined as the amplitude at the negative-phase inertial frequency, i.e., $-f_0$, after the complex Fourier transformation. We note that the energies contributed to the amplitude at $-f_0$ comprise the potential contributions from quasi-semidiurnal inertial and tidal oscillations, and the high-frequency components of wind and oceanic forcing; while that in the positive phase excludes contributions from inertial oscillation, and only comprises other components compared to that in a negative phase. This is because the spectral peaks associated with the tidal oscillation are roughly symmetric at positive and negative phases as a first order approximation (Gimbert et al., 2012). On the contrary, the spectral peak associated with the inertial oscillation is asymmetric, and only occurs in the negative phase in the Arctic Ocean. Thus, we can identify the seasonal changes in the contributions of inertial oscillation by comparing the amplitude at the negative-phase quasi-semidiurnal frequency, i.e., IMI, with that in the positive phase (hereinafter referred to as positive-phase amplitude, short: PHA). Such method to separate the inertial oscillation from the tidal oscillation has been used by Lammert et al. (2009), who attempted to identify cyclone-induced inertial ice oscillation in Fram Strait. The background noise originating from high-frequency components of wind and oceanic forcing can shift the local maximums slightly from the targeted frequencies of IMI and PHA (Geiger and Perovich, 2008). Thus, we identify the local maximum amplitude in the range of $-f_0 \pm 0.03$ for the IMI and in the range of $2 \pm 0.03$ for the PHA. From artificial identification, such ranges can ensure almost all quasi-semidiurnal signals won't be missed. If no local maximum can be identified within the defined ranges, we use the amplitudes at $-f_0$ and 2 as the

IMI and PHA. Such situation is rare for the IMI, i.e., approximately with 15% cases; while it is prevalent for the PHA, i.e., approximately with 95% cases. This implies an inertial oscillation is prevalent, while the tidal oscillation can be ignored regardless of season and buoy under consideration, which might be related to the fact that all the buoys drifted over the deep waters beyond the continental shelf through the study period.

[revised manuscript text omitted]
 patterns of seasonal change patterns and the linearly dependence on the logarithm of the temporal scale are consistent as those obtained using the threshold of 15%.

Therefore, the understanding of the localization of the ice deformation derived from this study is not very sensitive to the selected threshold.

**3.4 Spatial differences in the trends of sea ice loss in the PAO and their implications for sea ice kinematics and deformation**

Summer iSea ice conditions in the melt season have profound effects on sea ice dynamic and thermodynamic processes in the following winters. For example, eEnhanced divergence of summer sea ice leads to increased absorption of solar radiation by the upper ocean and delays onset of ice growth (e.g., Lei et al., 2020b). As shown in Fig. 15, the long-term decrease of sea ice concentration in the first half of September, when Arctic sea ice extent reaches its annual minimum (Comiso et al., 2017), is stronger in the southern and western parts of the study region than in the north and the east. The western and southern parts of the study region have become ice free in September during recent years.

On the contrary, there is no significant trend in ice concentration in the first half of September along the trajectory of the easternmost buoy (Fig. 15e). This suggests that, the melting period is getting longer in the southern and western parts of the PAO compared to the north and east. Consequently, the spatial gradient of ice thickness in the PAO, especially during autumn and early winter, will be further enhanced through delaying the onset of ice growth and reducing ice thickness in the south and west. 
[revised manuscript text omitted]

---

## Author Response (AR2)

**Dear Editor,**

Please see our second-time revised manuscript entitled "**Seasonal changes in sea ice kinematics and deformation in the Pacific Sector of the Arctic Ocean in 2018/19**" [Paper # tc-2020-211] to The Cryosphere. According to the comments from the anonymous reviewer, we made a minor revision of the manuscript by carrying out the following tasks:

1)  We added additional calculation and discussion of the high-resolution temporal changes in inertial signals based on the 5-day time-window. Those materials are helpful to compare the difference of seasonal attenuation of inertial signals of sea ice motion among various regions.
2)  In the conclusion, we added the discussion on the limitations of our study.
3)  We further improved language of the manuscript.

Please find the following files in our submission package:
1) The clean manuscript, 2) the manuscript with tracked changes, and 3) the response letter.

Thank you for your time.

Sincerely,
Ruibo Lei and co-authors

**Reply to reviewer**

1 In general I am happy with the response to my review. As regards the signal to noise ratio, I still think this is an important topic of discussion that would be easier to have if some figures were clearer in the manuscript. In particular figure 9 does not have sufficient resolution to resolve the tidal/inertial peak.

We supplemented the calculation and discussion on the high-resolution temporal changes in inertial signals based on the temporal window of 5 days (Lines 405-419 and Figure 10, the line number refer to the revised manuscript with track changes), which is helpful to compare the difference of seasonal attenuation of inertial signals of sea ice motion among various regions and identify the potential peaks of tidal and inertial signals.

We would still prefer to keep the original Figure 9, because it can give a full picture of spatiotemporal variations of the semidiurnal signal of sea ice motion.

2 The significance of trends in $\beta$ and $c$ are also not so clear. It could be argued this is also the case in previous papers on this topic.

We believe this unclarity might be contributed by the complicity of ice field and limitation of our study. Therefore, in the conclusion, we added the discussions on the limitations of our study, such as the limited scale range of buoy array. Some suggestions have been given for the future plans of observation and study of Arctic sea ice deformation. (Lines 611-633)

3 There are a few places in the new text where grammar needs small adjustments. Make sure you proof read before publication. Overall the English is very understandable, but do check grammar.

We checked the language and grammar through the manuscript, and made the expressions clearer.